# On the Effect of Positional Encoding for In-context Learning in Transformers

## Abstract

Transformer models have demonstrated a remarkable ability to perform a wide range of tasks through in-context learning (ICL), where the model infers patterns from a small number of example prompts provided during inference. However, empirical studies have shown that the effectiveness of ICL can be significantly influenced by the order in which these prompts are presented. Despite its significance, this phenomenon has been largely unexplored from a theoretical perspective. In this paper, we theoretically investigate how positional encoding (PE) affects the ICL capabilities of Transformer models, particularly in tasks where prompt order plays a crucial role. We examine two distinct cases: linear regression, which represents an order-equivariant task, and dynamic systems, a classic time-series task that is inherently sensitive to the order of input prompts. Theoretically, we evaluated the change in the model output when positional encoding (PE) is incorporated and the prompt order is altered. We proved that the magnitude of this change follows a convergence rate of $\mathcal{O}(k/N)$, where $k$ is the degree of permutation to the original prompt and $N$ is the number of in-context examples. Furthermore, for dynamical systems, we demonstrated that PE enables the Transformer to perform approximate gradient descent (GD) on permuted prompts, thereby ensuring robustness to changes in prompt order. These theoretical findings are experimentally validated.

## 1 Introduction

Large language models (LLMs) have shown remarkable in-context learning (ICL) capabilities (Brown et al., 2020). When provided with a few prompts as examples, these models can accurately predict outcomes for new tasks without requiring any parameter updates. This intriguing ability has sparked significant interest, prompting a recent wave of research aimed at developing a white-box theoretical understanding of ICL (Xie et al., 2022; Akyürek et al., 2022; Von Oswald et al., 2023; Ahn et al., 2023; Wu et al., 2023; Guo et al., 2023; Bai et al., 2024; Wang et al., 2024).

Despite the advantages of this remarkable phenomenon, Lu et al. (2021) found that the ICL ability of LLMs, such as GPT-3, is highly sensitive to prompt order. This sensitivity can result in performance ranging from near state-of-the-art to almost random guessing, depending on the prompt arrangement. This finding is surprising given that the Transformer architecture (Vaswani, 2017) is inherently permutation invariant (Yun et al., 2019; Lee et al., 2019), suggesting that changing the order of prompts should not affect the model's output. However, positional encoding (PE), a mechanism designed to incorporate order information into the otherwise permutation-invariant architecture, disrupts this invariance. Since the introduction of the Transformer, numerous PE variants (Vaswani, 2017; Brown et al., 2020; Zhang et al., 2022; Chowdhery et al., 2023; Touvron et al., 2023; Le Scao et al., 2023; Su et al., 2024) have been developed . While positional encoding has been extensively studied in tasks such as language modeling and machine translation, its specific impact on ICL remains an underexplored area of research.

In this paper, we aim to investigate how PE affects the ICL capabilities of Transformer models. We focus on two representative cases: linear regression, an order-insensitive task, and a simple dynamical system, which is highly order-sensitive. Our findings indicate that positional encoding (PE) does not statistically harm ICL performance in order-invariant tasks, such as linear regression, while enhancing order robustness in order-sensitive tasks, such as dynamical systems.

The key contributions of this paper are summarized as follows:

- We provide a sufficient condition on the weight matrices of the Transformer (section 3.1) that ensures it maintains permutation invariance, regardless of the input task type.

- For linear regression tasks, where predictions are ideally invariant to prompt order, we theoretically demonstrate that the change in ICL predictions caused by prompt order shifts is bounded by $\mathcal{O}(k/N) \cdot \epsilon$ for Transformers with positional encoding (PE) (section 3.2). This indicates minimal impact on ICL performance by PE.

- For dynamical systems, which closely resemble natural language processing tasks, we show that the theoretical bounds are consistent with those for linear regression, highlighting robustness to prompt order changes. Moreover, we find that PE enables Transformers to perform approximate gradient descent (GD) on permuted prompts, further proving the robustness of Transformers' ICL capabilities (section 3.3).

- We also validate our theoretical findings through experiments (section 4). The absolute differences in outputs between differently ordered prompts closely match our theoretical predictions (figs. 1 and 4) for both linear regression and dynamical systems.

## 2 PRELIMINARIES

### 2.1 TRANSFORMERS

A Transformer layer contains two sub-layers, the attention layer and the MLP layer. We denote the input sequence to the transformer as $\boldsymbol{h} = [\boldsymbol{h}_1, \cdots, \boldsymbol{h}_N] \in \mathbb{R}^{D \times N}$.

**Definition 2.1.** *(Attention layer) An attention layer with $M$ heads is denoted as* $\mathrm{Attn}_{\boldsymbol{\theta}}(\cdot)$, *where* $\theta = \{V_m, Q_m, K_m\}_{m \in [M]}$. *The output of this layer on the input matrix $H$ is:*

$$\mathrm{Attn}_{\boldsymbol{\theta}}(H) = H + \frac{1}{N} \sum_{m=1}^{M} (V_m H) \times \bar{\sigma}((Q_m H)^{\top}(K_m H)),$$

*where $\bar{\sigma} : \mathbb{R} \to \mathbb{R}$ is an activation function . For each column, we denote $\boldsymbol{h}_i^+ := [\mathrm{Attn}_{\boldsymbol{\theta}}(H)]_i$ and get:*

$$\boldsymbol{h}_i^+ = \boldsymbol{h}_i + \frac{1}{N} \sum_{m=1}^{M} \sum_{j=1}^{N} \bar{\sigma}\left(\langle Q_m \boldsymbol{h}_i, K_m \boldsymbol{h}_j \rangle\right) V_m \boldsymbol{h}_j.$$

In this paper we consider the linear self attention (LSA) layer following previous works (Dai et al., 2023; Mahankali et al., 2023; Ahn et al., 2023; Von Oswald et al., 2023). The attention layer is followed by the MLP layer.

**Definition 2.2.** *(MLP layer) An MLP layer with hidden dimension $D'$ is denoted as* $\mathrm{MLP}_{\boldsymbol{\theta}}(\cdot)$, *where $\boldsymbol{\theta} = (W_1, W_2) \in \mathbb{R}^{D' \times D} \times \mathbb{R}^{D \times D'}$. The output of this layer on input $\boldsymbol{h}$ is*

$$\mathrm{MLP}_{\boldsymbol{\theta}}(H) = H + W_2 \, \sigma(W_1 H),$$

*where $\sigma : \mathbb{R} \to \mathbb{R}$ is an activation function. For each column:*

$$[\mathrm{MLP}_{\boldsymbol{\theta}}(\boldsymbol{h})]_i = \boldsymbol{h}_i + W_2 \, \sigma(W_1 \boldsymbol{h}_i).$$

In the MLP layer $\sigma(t) = \max\{t, 0\}$ is the ReLU activation. A Transformer layer (block) contains one attention layer followed by one MLP layer, and we denote it as $\mathrm{TF}_i$, with $i$ indicating the $i$-th layer of a Transformer. We also denote a full Transformer by TF without specifying the number of layers.

### 2.2 POSITIONAL ENCODING

**Absolute positional encoding.** The absolute positional encoding (APE) which is used in the original Transformer paper (Vaswani, 2017), where positional encoding vectors ($\boldsymbol{p}_i$) are added to the corresponding word embeddings, resulting in a new hidden state at position $i$:

$$\boldsymbol{h}'_i = \boldsymbol{h}_i + \boldsymbol{p}_i.$$

Throughout this paper, we consider the one-hot positional encoding which allows precise study of how prompt order changes affect predictions, independent of complex encoding schemes. The one-hot PE is of the form

$$\boldsymbol{p}_i = \begin{bmatrix} \mathbf{0}_{i-1} \\ 1 \\ \mathbf{0}_{N-i} \end{bmatrix} \in \mathbb{R}^N,$$

where $N$ is the number of columns of the input matrix. We also concatenate the positional encoding with the input matrix $H$ instead of adding it directly to $H$ (see eq. (1)), following previous works (Guo et al., 2023; Bai et al., 2024; Wang et al., 2024).

**Rotary positional embedding.** Rotary Positional Encoding (RoPE) (Su et al., 2024) is an alternative positional encoding scheme that incorporates relative position information through rotation operations in the vector space. Unlike absolute positional encodings that are added to or concatenated with token embeddings, RoPE encodes positional information by rotating the query and key vectors in the self-attention mechanism using rotation matrices. The rotation matrix $\mathcal{R}_m$ is a block-diagonal matrix defined as:

$$\mathcal{R}_m = \begin{bmatrix} \cos(m\theta_0) & -\sin(m\theta_0) & 0 & 0 & \cdots & 0 & 0 \\ \sin(m\theta_0) & \cos(m\theta_0) & 0 & 0 & \cdots & 0 & 0 \\ 0 & 0 & \cos(m\theta_1) & -\sin(m\theta_1) & \cdots & 0 & 0 \\ 0 & 0 & \sin(m\theta_1) & \cos(m\theta_1) & \cdots & 0 & 0 \\ \vdots & \vdots & \vdots & \vdots & \ddots & \vdots & \vdots \\ 0 & 0 & 0 & 0 & \cdots & \cos(m\theta_{d/2-1}) & -\sin(m\theta_{d/2-1}) \\ 0 & 0 & 0 & 0 & \cdots & \sin(m\theta_{d/2-1}) & \cos(m\theta_{d/2-1}) \end{bmatrix}$$

where $\theta_i = 10000^{-2i/d}$ for $i = 0, 1, \ldots, d/2 - 1$, and $d$ is the dimension of the query/key vectors. For each dimension pair $(2i, 2i + 1)$ in the query/key vectors, we apply the rotation:

$$\begin{bmatrix} q_{2i}^m \\ q_{2i+1}^m \end{bmatrix} = \begin{bmatrix} \cos m\theta_i & -\sin m\theta_i \\ \sin m\theta_i & \cos m\theta_i \end{bmatrix} \begin{bmatrix} q_{2i} \\ q_{2i+1} \end{bmatrix}.$$

The attention score between position $m$ and $n$ becomes:

$$a_{m,n} = \frac{(\mathcal{R}_m Q \boldsymbol{h}_m)^\top (\mathcal{R}_n K \boldsymbol{h}_n)}{\sqrt{d_k}} = \frac{\boldsymbol{h}_m^\top Q^\top \mathcal{R}_{n-m} K \boldsymbol{h}_n}{\sqrt{d_k}},$$

where $\mathcal{R}_{n-m}$ is the block-diagonal rotation matrix for relative position $n - m$. This formulation ensures that the attention scores between queries and keys depend only on their relative distance m-n, providing inherent relative position awareness in the self-attention computation.

## 2.3 IN-CONTEXT LEARNING

A complete in-context learning (ICL) process contains two stages: pretraining and inference. In the pretraining stage, a Transformer is trained on meta-data generated from $n$ different tasks, where each data point $(\boldsymbol{x}, y)$ is sampled from a task-specific distribution $\mathrm{P}_i$, where $i = 1, \cdots, n$ indexes the tasks. During the inference stage, the prompts are sampled from a distribution $\mathrm{P}'_k$ corresponding to task $k$. Here $\mathrm{P}'_k$ during inference can differ from $\mathrm{P}_k$ in pretraining. For example, let the $k$-th task denote a linear regression problem $y = \boldsymbol{w}^\top \boldsymbol{x}$, the weight $\boldsymbol{w}_{\text{pretrain}}$ used during pretraining could be different from $\boldsymbol{w}_{\text{inference}}$ used during inference. We denote the prompts consisting of in-context examples as $\mathcal{D} = (\boldsymbol{x}_i, y_i)_{i \in [N]}$, representing $N$ examples sampled from the task distribution. A novel input $\boldsymbol{x}_{N+1}$ is sampled from $\mathrm{P}_x$, forming the input to the Transformer as a pair $(\mathcal{D}, \boldsymbol{x}_{N+1})$. Here $\boldsymbol{x}_i \in \mathbb{R}^d, y_i \in \mathbb{R}$.

More specifically, we denote the input to the transformer as

$$H = \begin{bmatrix} \boldsymbol{x}_1 & \boldsymbol{x}_2 & \cdots & \boldsymbol{x}_N & \boldsymbol{x}_{N+1} \\ y_1 & y_2 & \cdots & y_N & 0 \\ \boldsymbol{p}_1 & \boldsymbol{p}_2 & \cdots & \boldsymbol{p}_N & \boldsymbol{p}_{N+1} \\ \mathbf{0} & \mathbf{0} & \cdots & \mathbf{0} & \mathbf{0} \end{bmatrix} \in \mathbb{R}^{D \times (N+1)}, \tag{1}$$

where $\boldsymbol{p}_i \in \mathbb{R}^{N+1}$ is the one-hot positional encoding, and $\boldsymbol{0} \in \mathbb{R}^{D-N-d-2}$ is the zero padding. As mentioned in section 2.2, here we concatenated the positional encoding with the input matrix, rather than adding them, to highlight the impact of positional encoding while preserving its fundamental characteristics.

A Transformer processes the input prompt $H$ and generates a prediction for the label corresponding to $\boldsymbol{x}_{N+1}$. The prediction value $\hat{y}_{N+1}$ is stored in the output matrix $\tilde{H}$ at the position immediately following $y_N$. We say in-context learning succeeds if $\hat{y}_{N+1}$ and $y_N$ is close enough, or $\epsilon$-close, under a certain metric associated with task $k$ (In this work we set the metric as the MSE loss).

## 3 MAIN RESULTS

In this section, we first provide a high level approach towards understanding how the positional encoding could maintain the permutation invariance of Transformers. Then we examine two types of in-context learning (ICL) tasks: linear regression and first-order difference equations. Linear regression, a well-established ICL task extensively studied in prior works (Bai et al., 2024; Wang et al., 2024), serves as a lens to explore the underlying mechanisms of Transformers' ICL capabilities. This task is permutation invariant, so the order of prompts does not influence the predictions. In contrast, first-order difference equations, a time-series task studied by Li et al. (2023); Guo et al. (2023), are highly sensitive to prompt order, making them an ideal test case for assessing the effectiveness of positional encoding in ICL.

### 3.1 HIGH LEVEL APPROACH

The objective is to analyze how positional encoding affects the output of a Transformer when the input prompts are permuted. To formalize this, we first define the raw input matrix (without concatenated positional encoding) as

$$H' = \begin{bmatrix} \boldsymbol{x}_1 & \boldsymbol{x}_2 & \cdots & \boldsymbol{x}_N & \boldsymbol{x}_{N+1} \\ y_1 & y_2 & \cdots & y_N & 0 \\ \boldsymbol{0}_{D-d-1} & \boldsymbol{0}_{D-d-1} & \cdots & \boldsymbol{0}_{D-d-1} & \boldsymbol{0}_{D-d-1} \end{bmatrix},$$

where $\{\boldsymbol{x}_i\}_{i \in [N+1]} \in \mathbb{R}^d$ are the feature vectors, $\{y_i\}_{i \in [N+1]} \in \mathbb{R}$ are the corresponding labels, and $\boldsymbol{0}_{D-d-1}$ represents zero-padding to align the dimensions. The positional encoding matrix is defined as:

$$E = \begin{bmatrix} \boldsymbol{0}_{d+1} & \cdots & \boldsymbol{0}_{d+1} & \boldsymbol{0}_{d+1} \\ \boldsymbol{p}_1 & \cdots & \boldsymbol{p}_N & \boldsymbol{p}_{N+1} \\ \boldsymbol{0}_{D-d-N-2} & \cdots & \boldsymbol{0}_{D-d-N-2} & \boldsymbol{0}_{D-d-N-2} \end{bmatrix},$$

where $\{\boldsymbol{p}_i\}_{i \in [N+1]} \in \mathbb{R}^{N+1}$ are the positional encoding vectors. Consequently, the full input to the Transformer becomes $H = H' + E$.

#### 3.1.1 POSITIONAL ENCODING AFFECTS ATTENTION OUTPUT

For the attention layer, let the attention operation be denoted by $f := \text{Attn}$. If $P$ is any permutation matrix, it is known that $f(H'P) = f(H')P$. When positional encoding is added, the difference between the attention outputs becomes

$$f(H'P + E) - f(H' + E)$$
$$= f(H'P + E) - f(H'P + EP) + f((H' + E)P) - f(H' + E)$$
$$= f(H'P + E) - f(H'P + EP).$$

Denote $g_A(B) = f(A + B) - f(A)$, then we can rewrite the above equation as

$$f(H'P + E) - f(H' + E)$$
$$= f(H'P + E) - f(H'P + EP)$$
$$= f(H'P + E) - f(H'P) - (f(H'P + EP) - f(H'P))$$
$$= g_{H'P}(E) - g_{H'P}(EP)$$
$$= (g_{H'P}(E) - g_{H'P}(E)P) + (g_{H'P}(E)P - g_{H'P}(EP)).$$

Since we are interested only in the last column of the output, the first term vanishes since the permutation matrix $P$ doesn't affect the last column, so we only need to study $g_{H'P}(E)P - g_{H'P}(EP)$. This implies that in the presence of positional encoding, the effect of permutation on the Transformer output depends on whether the function $g$ is permutation invariant. Expanding the definition of $f$, we find that

$$g_A(B) \approx B + \frac{1}{N} \sum_{m=1}^{M} V_m(AB^\top R_m A + BA^\top R_m A + AA^\top R_m B),$$

where $R_m = Q_m^\top K_m$. Next we compute

$$g_A(BP) - g_A(B)P \approx \frac{1}{N} \sum_{m=1}^{M} V_m(A(P^\top B^\top R_m A - B^\top R_m AP) + B(PA^\top R_m A - A^\top R_m AP)),$$

where we omit the higher order terms of $B$. This difference term is generally non-zero, indicating that positional encoding impacts the attention output and compromises its permutation invariance. While this property hinders performance on permutation-invariant tasks like in-context linear regression (where input order should be irrelevant), it could potentially be beneficial for tasks where sequence ordering carries meaningful information, such as time-series prediction or language modeling. In the following, we first demonstrate that under a specific assumption, permutation invariance can still be preserved.

### 3.1.2 Attention Layer Preserves Permutation Invariance

By substituting $A = H'P$ and $B = E$, a sufficient condition for $g$ to be permutation invariant is

**Condition 3.1.** $R_m$ *is a symmetric matrix of the form*

$$R_m = \left[ \begin{array}{c|c} \begin{array}{c} S_m \\ \mathbf{0} \\ T_m \end{array} & U_m \end{array} \right],$$

*with the dimension of block matrices satisfying:* $S_m \in \mathbb{R}^{(d+1)\times(d+1)}, \mathbf{0} \in \mathbb{R}^{(N+1)\times(d+1)}, T_m \in \mathbb{R}^{(D-N-d-2)\times(d+1)}$ *and* $U_m \in \mathbb{R}^{D\times(D-d-1)}$.

When Condition 3.1 holds, it follows that $B^\top R_m A = \mathbf{0} \in \mathbb{R}^{(N+1)\times(N+1)}$, and $A^\top R_m A$ becomes symmetric. This symmetry ensures that $PA^\top R_m A = A^\top R_m AP$, thereby preserving the permutation invariance of $g$. The above condition can be further loosened if we don't require both terms in the decomposition of $g_A(BP) - g_A(B)P$ to be zero matrices for each head.

**Remark 3.1.** *If the positional encoding is not one-hot, Condition 3.1 should be tightened to require* $\mathbf{0} \in \mathbb{R}^{(N+1)\times D}$. *Note that the matrix* $R_m \in \mathbb{R}^{D\times D}$, *so this is a rather strong restriction, especially when $N$ is large.*

### 3.1.3 MLP Layer Preserves Permutation Invariance

Let $\phi$ denote the MLP layer, then

$$\phi(H) = H + W_2\sigma(W_1 H).$$

Similarly we can compute how the positional encoding affects the output of the MLP layer.

$$\phi(H'P + E) - \phi(H' + E) = H' - H + W_2(\sigma(W_1(H'P + E)) - \sigma(W_1(H' + E))),$$

where the first term need not be considered provided that the permutation $P$ doesn't affect the the last column in $H$. By the property of $\sigma$ and the structure of $H', E$, we have $\sigma(W_1(H'P + E)) - \sigma(W_1(H' + E)) = \sigma(W_1 H')P - \sigma(W_1 H')$, thus the last column is also unaffected. This shows that the MLP layer still maintains permutation invariance after adding the positional encoding.

Now we summarize the result reached so far as:

**Proposition 1.** *There exists pretrained Transformers (satisfying Condition 3.1), such that positional encoding does not compromise the permutation-invariance property of Transformers.*

The proposition implies that positional encoding can interfere with the Transformer architecture's inherent permutation invariance. This disruption presents challenges when applying Transformers to permutation-invariant in-context learning (ICL) tasks. However, the findings suggest that specialized pretraining on such tasks may enable the model to compensate for these effects. Specifically, a Transformer pretrained on permutation-invariant ICL tasks could potentially learn to overcome the limitations introduced by positional encoding, effectively mitigating its adverse impacts on model performance.

## 3.2 PE EFFECT ON LINEAR REGRESSION

In this section we consider linear regression tasks, which is the most common setting in ICL analysis studied by many (Akyürek et al., 2022; Von Oswald et al., 2023; Ahn et al., 2023; Wu et al., 2023; Gatmiry et al., 2024).

### 3.2.1 ONE-HOT PE

We first state a mild condition which bounds the Transformer's weight matrices.

**Assumption 3.1.** *Consider a transformer pretrained on a task $y = f(\boldsymbol{x})$, where $\boldsymbol{x} \in \mathbb{R}^d$, with $N$ in-context examples in each data point. The pretrained Transformer satisfies*

$$\max\{| \left(Q^\top K\right)_{d+N+1,d+2} - \left(Q^\top K\right)_{d+N+1,d+3} |, |V_{d+1,d+2} - V_{d+1,d+3}|\} \leq \epsilon,$$

*where $\epsilon$ is a small quantity.*

The assumption only requires two elements in the matrices $Q^\top K$ and $V$ to be close enough, which is a rather loose assumption in that it doesn't require the specific $Q, K, V$ construction in previous works (Von Oswald et al., 2023; Li et al., 2023; Ahn et al., 2023; Wang et al., 2024; Bai et al., 2024).

**Theorem 3.1.** *Under Assumption 3.1, assume that each element of $\boldsymbol{x}_i$, denoted as $\boldsymbol{x}_i^k$, follows a normal distribution $\mathcal{N}(0, 1/2)$. For linear regression tasks $y_i = \boldsymbol{w}^\top \boldsymbol{x}_i$, let $\Delta y_{N+1} = \hat{y}_{N+1} - y_{N+1}$, where $\hat{y}_{N+1}$ represents a Transformer block's prediction after applying a $k$-degree permutation to the prompt, and $y_{N+1}$ is the prediction based on the original prompt. Then, the following result holds:*

$$\sup \mathbb{E}[|\Delta y_{N+1}|] \longrightarrow C_1 \frac{k\sqrt{d}}{N}\epsilon + \frac{2k}{N}\epsilon^2 \ (d, N \rightarrow \infty),$$

*where $C_1$ is a constant that depends on $\boldsymbol{x}_{N+1}, Q, K, V$.*

The core proof techniques include transforming the change in the output position caused by a permutation into a random variable with well-defined statistical properties and leveraging group theory to systematically extend the result for a single transposition to the general case of a $k$-degree permutation. The detailed proof is in Appendix B. To the best of our knowledge, this is the first formal result that explicitly demonstrates how positional encoding influences the Transformer's output in the context of in-context learning (ICL) predictions.

**Remark 3.2.** *Our analysis introduces two key innovations in understanding Transformers' permutation sensitivity. First, we develop a novel probabilistic framework that characterizes positional encoding effects by modeling permutation-induced output changes as random variables with provable statistical properties. Second, we employ group-theoretic techniques to generalize from single transpositions to arbitrary k-degree permutations, establishing a complete theoretical characterization. To the best of our knowledge, this approach yields the first formal proof (section B) quantifying how positional encoding systematically affects Transformer outputs in in-context learning scenarios.*

The previous theorem is the result for a single Transformer layer with only one attention head. Now we provide the result for a more general multiple attention head, $L$-layer setting.

> **Corollary 3.1.** *There exist pretrained L-layer Transformers for which the difference bound in Theorem 3.1 remains valid, up to a factor of L.*

**Remark 3.3.** *Although Corollary 3.1 seems like a more general version of Theorem 3.1, it actually requires stricter conditions on the Transformer weight matrices ($\epsilon$ must be 0 in Assumption 3.1) to maintain the same input format.*

### 3.2.2 RoPE

For the widely used Rotary Position Embedding (RoPE), we derive a theorem parallel to Theorem 3.1 by utilizing the notation in section 2.2.

> **Theorem 3.2.** *Assume that each element of $\boldsymbol{x}_i$, denoted as $\boldsymbol{x}_i^k$, follows a normal distribution $\mathcal{N}(0, 1/2)$. For linear regression tasks $y_i = \boldsymbol{w}^\top \boldsymbol{x}_i$, let $\Delta y_{N+1} = \hat{y}_{N+1} - y_{N+1}$, where $\hat{y}_{N+1}$ represents a Transformer block's prediction using RoPE after applying a $k$-degree permutation to the prompt, and $y_{N+1}$ is the prediction based on the original prompt. Then, the following result holds:*
>
> $$\sup \mathbb{E}[|\Delta y_{N+1}|] \longrightarrow C_{RoPE}\frac{kd^3}{N},$$
>
> *where $C_{RoPE}$ is a constant that depends on $\boldsymbol{x}_{N+1}, Q, K, V, \Theta = \{\theta_i\}_i$.*

The proof is deferred to section B. Theorem 3.2 doesn't rely on any assumptions for the weight parameter of the transformer, this is because the rotary embedding doesn't intervene with the hidden dimension. However, the reliance on the input dimension $d$ does grow to $d^3$ due to the dimension dependent Frobenius norm of the rotation matrix $\mathcal{R}_n$. Unlike absolute positional encodings, RoPE's rotational structure introduces additional dimensional dependencies through the pairwise rotation operations across embedding dimensions.

**Remark 3.4.** *Our analysis follows a similar technical framework to Theorem 3.1, with the key distinction lying in how RoPE modulates attention scores between permuted input columns. Theorem 3.2 reveals that, compared to one-hot positional encoding, transformers utilizing RoPE exhibit heightened sensitivity to the input dimension $d$. This manifests as a $d^3$ scaling factor in the error bound, suggesting that RoPE-based models may experience greater instability in in-context learning performance, particularly in high-dimensional settings.*

### 3.3 PE EFFECT ON FIRST ORDER DIFFERENCE EQUATIONS

We consider the first order difference equation in this section. This is a more realistic setting since modern large language models are next-token predictors, and the dynamic of the first order difference equation resembles the essence of the next-token predicition pattern.

For this scenerio we consider the input to the Transformer as:

$$H = \begin{bmatrix} \boldsymbol{x}_1 & \boldsymbol{x}_2 & \cdots & \boldsymbol{x}_N & \boldsymbol{x}_{N+1} \\ 0 & 0 & \cdots & 0 & 0 \\ \boldsymbol{p}_1 & \boldsymbol{p}_2 & \cdots & \boldsymbol{p}_N & \boldsymbol{p}_{N+1} \end{bmatrix} \in \mathbb{R}^{D \times (N+1)}, \tag{2}$$

where $p_i$ is the one-hot positional encoding, and

$$\boldsymbol{x}_{i+1} = A\boldsymbol{x}_i + \boldsymbol{b}.$$

**Theorem 3.3.** *Under Assumption 3.1, assume $\boldsymbol{x}_0^k \sim \mathcal{N}(0, 1/2)$. For first order difference equation $\boldsymbol{x}_{i+1} = A\boldsymbol{x}_i + \boldsymbol{b}$, define $\Delta \boldsymbol{x}_{N+1} = \hat{\boldsymbol{x}}_{N+1} - \boldsymbol{x}_{N+1}$, where $\hat{\boldsymbol{x}}_{N+1}$ represents the transformer's prediction after applying a k-degree permutation to the prompt, and $\boldsymbol{x}_{N+1}$ corresponds to the prediction based on the original prompt. Then the following result holds:*

$$\sup \mathbb{E}[\|\Delta \boldsymbol{x}_{N+1}\|_2] \longrightarrow C_2 \frac{kd}{N}\epsilon + \frac{2k\sqrt{d}}{N}\epsilon^2 \ (d, N \to \infty),$$

*where $C_2$ is a constant dependent on $\boldsymbol{x}_{N+1}, Q, K, V$.*

It is important to note that Theorem 3.3 demonstrates the stability of positional encoding's effect on the shifted prompt and suggests that prediction accuracy could remain comparable to the original prompt. However, it does not explain why positional encoding might improve the robustness of a Transformer to changes in prompt order compared to the scenario without positional encoding. Building on the findings of Guo et al. (2023), we derive the ICL prediction error for a Transformer learning the dynamics system.

**Lemma 3.1.** *For any $\epsilon > 0$, there exists a Transformer with $\mathcal{O}(\epsilon^{-1})$ blocks such that for the input $\tilde{H}$ of the form*

$$\tilde{H} = \begin{bmatrix} \boldsymbol{x}_1 & \boldsymbol{x}_2 & \cdots & \boldsymbol{x}_N & \boldsymbol{x}_{N+1} \\ \boldsymbol{0}_d & \boldsymbol{0}_d & \cdots & \boldsymbol{0}_d & \boldsymbol{0}_d \\ \boldsymbol{0}_d & \boldsymbol{x}_1 & \cdots & \boldsymbol{x}_{N-1} & \boldsymbol{x}_N \\ \boldsymbol{0}_d & \boldsymbol{x}_2 & \cdots & \boldsymbol{x}_N & \boldsymbol{x}_{N+1} \\ \boldsymbol{p}_1 & \boldsymbol{p}_2 & \cdots & \boldsymbol{p}_N & \boldsymbol{p}_{N+1} \end{bmatrix},$$

*the prediction of the Transformer $\hat{\boldsymbol{y}}_i = [\text{TF}(\tilde{H})]_{(d+1):2d, i}(i \in [N+1])$ satisfies*

$$\|\hat{\boldsymbol{y}}_i - A\boldsymbol{x}_i\|_2 \leq \sqrt{d}\epsilon,$$

*with d being the dimension of $\boldsymbol{x}$.*

Lemma 3.1 provides a bound on the error of the ICL output prediction based on a specific input format. By utilizing the above lemma, we get

**Theorem 3.4.** *For any $\epsilon > 0$, there exists a Transformer with $\mathcal{O}(\epsilon^{-1})$ layers such that for an input structured as described in eq. (2), it implements approximate GD on the input with shifted prompt order and the prediction for $\boldsymbol{x}_i$ ($i \in [N]$) satisfies the following upper bound:*

$$\|\hat{\boldsymbol{x}}_{i+1} - A\boldsymbol{x}_i\|_2 \leq (\sqrt{k}d + \sqrt{d})\epsilon,$$

*where $k, d$ represents the degree of permutation and the dimension of $\boldsymbol{x}$ respectively.*

This demonstrates that, despite input permutations, a Transformer with positional encoding can still perform in-context learning with a certain level of accuracy.

## 4 EXPERIMENTS

We conduct experiments on the two settings discussed in section 3, namely linear regression and first order difference equation. We pretrained several 12-layer, 8-head encoder transformer models with hidden space $D_{\text{hid}} = 256$, following settings in previous works (Garg et al., 2022; Li et al., 2023; Bai et al., 2024; Guo et al., 2023). We used ADAM optimizer with a learning rate of 1e-4. For linear regression, the data points are sampled from $\boldsymbol{x} \sim \mathcal{N}(\boldsymbol{0}, \boldsymbol{I}_d), \boldsymbol{w} \sim \mathcal{N}(\boldsymbol{0}, \boldsymbol{I}_d)$, where $d = 20$; for first order difference equation, $\{\boldsymbol{x}_i\}_{i \in [N+1]} \sim \mathcal{N}(\boldsymbol{0}, \boldsymbol{I}_d)$ and $\{A_j\}_{j \in [d]} \sim \mathcal{N}(\boldsymbol{0}, \boldsymbol{I}_d)$ ($A_j$ denotes the $j$-th row of $A$) with $\boldsymbol{b} \sim \mathcal{N}(\boldsymbol{0}, \boldsymbol{I}_d)$, where $d = 2$. $N$ denotes the number of in-context examples during pretraining and $N = 40$ for linear regression, $N = 10$ for first order difference equation.

These experiments show that the negative effect PE brings decays in an $\mathcal{O}(N^{-1})$ order and increases in an $\mathcal{O}(k)$ order (fig. 1), which strongly supports our theorem. What's more, PE is important in preserving the robustness of transformers in tackling order-sensitive ICL tasks such as dynamic systems (fig. 3), which also supports our theoretical findings.

**Linear Regression.** To evaluate in-context learning (ICL) performance on linear regression tasks, we pretrained two Transformer models (with or without PE). The models were trained with a batch size of 64 for 150,000 steps. During inference, we sampled 10,000 instances to estimate the expected mean squared error (MSE) loss.

We measured the expectation of the absolute difference in the output of a Transformer model with both one-hot PE and RoPE (figs. 1 and 2) pretrained with PE under increasing degrees of prompt permutation $k$. Specifically, the first $k$ prompts were flipped, and we sampled 10,000 instances over a batch size of 64 to approximate the expectation. The experimental results (figs. 1 and 2, left) showed that the increase in the absolute difference follows an order of $\mathcal{O}(k)$, consistent with our theoretical prediction in Theorems 3.1 and 3.2.

Next, we evaluated the effect of increasing the number of in-context examples while keeping the prompt permutation fixed (figs. 1 and 2 right). In this setup, we only swapped the order of the first two columns of the input matrix. The results demonstrated that the absolute difference decays at a rate of $\mathcal{O}(N^{-1})$, again matching our theoretical analysis in Theorems 3.1 and 3.2.

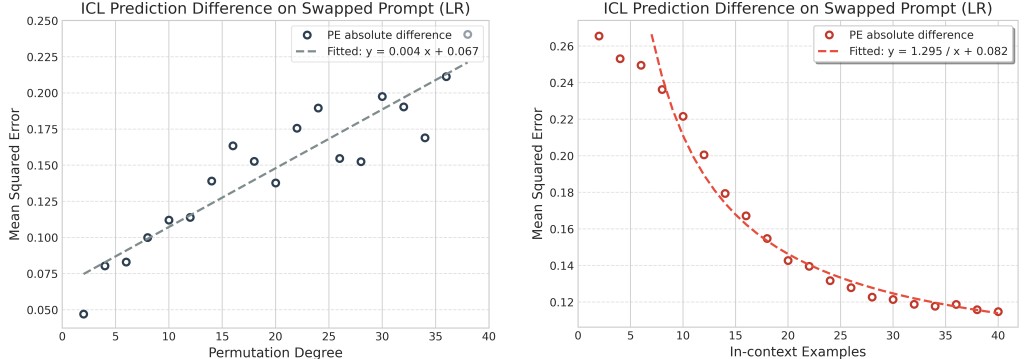

Figure 1: Experimental results on linear regression tasks. *Left*: The absolute difference of the prediction is proportional to the degree of permutation to the prompt. *Right*: The absolute difference of the prediction with swapped prompt order by a pretrained PE transformer can be fitted by an inverse proportional function.

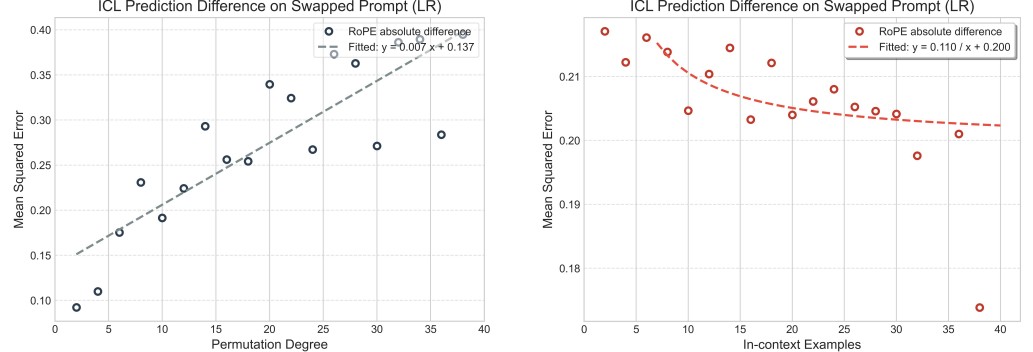

Figure 2: Experimental results on linear regression tasks for RoPE.

**First Order Difference Equation.** For the first order difference equations, we also pretrained two transformers following the experiment setting in the linear regression experiment. Note that the default number of in-context examples is 10 because the solution to the equation will converge to a constant quickly, resulting in the last few columns of the input matrix to be practically the

same. Therefore, too many in-context examples will make the transformers learn to merely copy the previous column during ICL inference, which is not intended. In fig. 3 left, as the number of in-context examples grows, the MSE loss tend to converge for both models. However, once the order of the in-context examples is swapped, fig. 3 right demonstrates that the performance of the model with PE is still robust but the model without PE predicts worse.

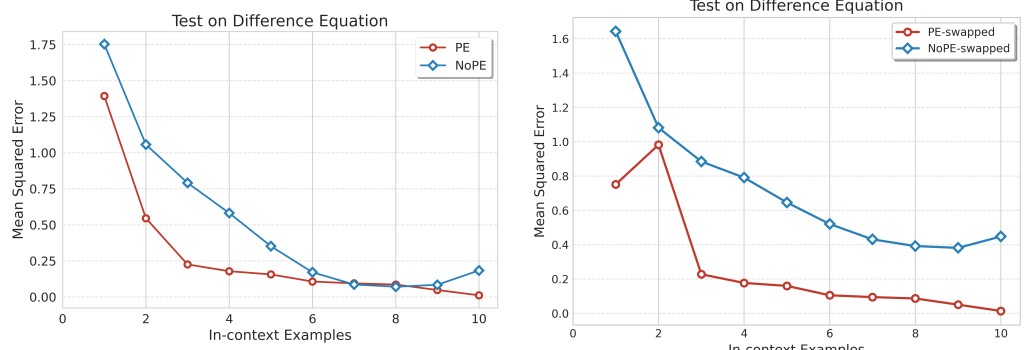

Figure 3: Experimental results on first order difference equation tasks. *Left*: The comparison of the ICL ability of two pretrained transformers (with or without PE). *Right*: The prediction with swapped prompt order by two pretrained transformers (with or without PE).

We also evaluated the absolute difference similiar the linear regression setting for increasing permutation degree $k$ (fig. 4 left) and increasing in-context example number $N$ (fig. 4 right), and the relationship between the MSE loss and $\mathcal{O}(N^{-1})$, $\mathcal{O}(k)$ matches Theorem 3.3.

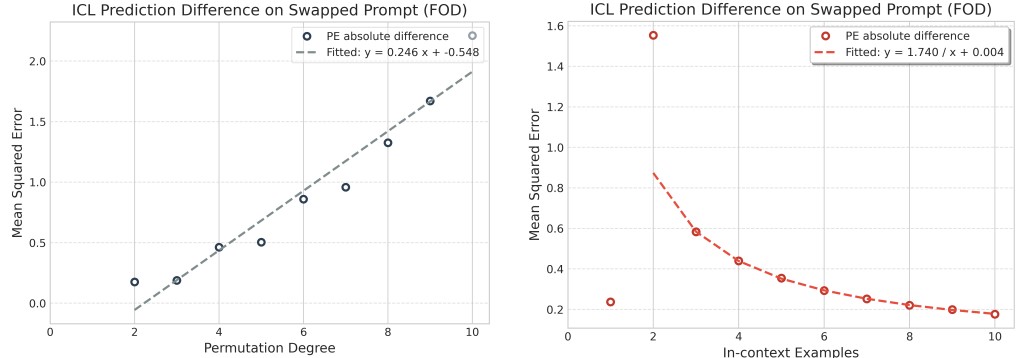

Figure 4: Experimental results on first order difference equation tasks. *Left*: The absolute difference of the prediction the fitted curve. *Right*: The absolute difference of the prediction with swapped prompt order by a pretrained PE transformer and the fitted curve.

## 5 CONCLUSION

This work provides both theoretical and empirical insights into how positional encoding influences the in-context learning (ICL) capabilities of Transformers on linear regression and dynamical systems tasks. For the linear regression task, we theoretically demonstrate that one-hot positional encoding can lead to instability in predictions with respect to prompt order. The prediction difference scales linearly with the permutation degree $k$ of the prompt, but diminishes at a rate of $\mathcal{O}(N^{-1})$ as the number of in-context examples $N$ increases. For the dynamical system task, we focus on a simple first-order difference equation, which mimics a natural language next-token prediction process with a context window size of one. Our theoretical analysis shows that the prediction difference caused by positional encoding follows the same order as in the linear regression task. Our Empirical results corroborate these theoretical findings, validating the predicted relationship between prompt order and prediction stability for both tasks.

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

## A  RELATED WORK

**In-context Learning.** In-context learning (ICL) has been studied both empirically and theoretically. Garg et al. (2022) empirically shows that Transformers can learn linear functions, two-layer ReLU neural networks, and decision trees in context. Min et al. (2022) studies what aspects of demonstrations impact the performance of ICL. As for the theoretical part, Xie et al. (2022) explains ICL as implicit Bayesian inference despite the difference between pretraining and inference distributions, while many other works (Akyürek et al., 2022; Von Oswald et al., 2023; Dai et al., 2023) interpret ICL as Transformers performing gradient descent (GD). These works mainly focus on linear models or their variants. Bai et al. (2024) investigates gradient descent on a wider range of functions, like 2-layer neural networks, and demonstrates the algorithm selection ability of Transformers. Wang et al. (2024) extends their work to $n$-layer neural network setting for more general function approximation. Besides those that study ICL mechanism, Lu et al. (2021) proves that order sensitivity is a common problem in ICL, which can result in performance ranging from near state-of-the-art to almost random guessing, depending on the prompt arrangement. To address this issue, Liu et al. (2021) and Liu et al. (2024) suggest arranging input in a particular way similar to curriculum learning to enhance in-context learning.

**Positional Encoding.** Vaswani (2017) proposed a sinusoidal positional encoding (PE) to capture the word order in the input. There are mainly two types of PEs: *absolute*, where positions are represented explicitly as numbers or vectors (e.g., $1, 2, 3, \dots$), or *relative*, where positional information is based on the distance between tokens (Kazemnejad et al., 2024). Below, we provide a brief overview of the common positional encoding methods used in Transformers.

*Absolute Position Embedding (APE)* assigns each position $i$ a position vector $\mathbf{p}_i$, which is added to the corresponding word embeddings. Non-parametric APE uses sinusoidal functions to generate embeddings for any position (Vaswani, 2017), while learned APE, as used in GPT-3 (Brown et al., 2020) and OPT (Zhang et al., 2022), trains position embeddings with the model parameters but cannot handle unseen positions, limiting the context window to a fixed length.

*T5's Relative Bias* maps the relative distance $(i - j)$ between tokens to a scalar bias $b = f(i - j)$ using a lookup table. This bias, learned during training, is added to the query-key dot product in the self-attention mechanism. Distances beyond a threshold are mapped to the same parameter to generalize to unseen distances.

*Rotary*, employed in models like PaLM (Chowdhery et al., 2023) and LLaMA (Touvron et al., 2023), applies a position-dependent rotation to the query and key representations before computing the attention dot product. This rotation ensures that the attention depends only on the relative distance between tokens, functioning as a form of relative positional encoding (Su et al., 2024).

*ALiBi*, utilized in BLOOM (Le Scao et al., 2023), subtracts a scalar bias from the attention score. This bias increases linearly with the distance between query and key tokens, introducing a recency bias that favors more recent tokens.

## B  PROOFS FOR SECTION 3.2

**Theorem 3.1.** *Under Assumption 3.1, assume that each element of $\boldsymbol{x}_i$, denoted as $\boldsymbol{x}_i^k$, follows a normal distribution $\mathcal{N}(0, 1/2)$. For linear regression tasks $y_i = \boldsymbol{w}^\top \boldsymbol{x}_i$, let $\Delta y_{N+1} = \hat{y}_{N+1} - y_{N+1}$, where $\hat{y}_{N+1}$ represents a Transformer block's prediction after applying a $k$-degree permutation to the prompt, and $y_{N+1}$ is the prediction based on the original prompt. Then, the following result holds:*

$$\sup \mathbb{E}[|\Delta y_{N+1}|] \longrightarrow C_1 \frac{k\sqrt{d}}{N}\epsilon + \frac{2k}{N}\epsilon^2 \ (d, N \to \infty),$$

*where $C_1$ is a constant that depends on $\boldsymbol{x}_{N+1}, Q, K, V$.*

*Proof.* We first consider the case where only two adjacent columns are permuted and then generalize to random permutation between $k$ columns.

**1. Permutation between two adjacent columns**

The original input is

$$
H_O = \begin{bmatrix} \boldsymbol{x}_1 & \boldsymbol{x}_2 & \cdots & \boldsymbol{x}_N & \boldsymbol{x}_{N+1} \\ y_1 & y_2 & \cdots & y_N & 0 \\ \boldsymbol{p}_1 & \boldsymbol{p}_2 & \cdots & \boldsymbol{p}_N & \boldsymbol{p}_{N+1} \\ \boldsymbol{0} & \boldsymbol{0} & \cdots & \boldsymbol{0} & \boldsymbol{0} \end{bmatrix} \in \mathbb{R}^{D \times (N+1)}.
$$

After the permutation (WLOG, we assume the permutation happens between column 1 and 2) the input becomes

$$
H_P = \begin{bmatrix} \boldsymbol{x}_2 & \boldsymbol{x}_1 & \cdots & \boldsymbol{x}_N & \boldsymbol{x}_{N+1} \\ y_2 & y_1 & \cdots & y_N & 0 \\ \boldsymbol{p}_1 & \boldsymbol{p}_2 & \cdots & \boldsymbol{p}_N & \boldsymbol{p}_{N+1} \\ \boldsymbol{0} & \boldsymbol{0} & \cdots & \boldsymbol{0} & \boldsymbol{0} \end{bmatrix} \in \mathbb{R}^{D \times (N+1)}.
$$

Now consider the impact of the permutation on the output of the last column $h_{N+1}$. From definition 2.1 we know that

$$
\boldsymbol{h}_{N+1}^+ = \boldsymbol{h}_{N+1} + \frac{1}{N+1} \sum_{m=1}^{M} \sum_{j=1}^{N+1} \sigma\left(\langle Q_m \boldsymbol{h}_{N+1}, K_m \boldsymbol{h}_j \rangle\right) V_m \boldsymbol{h}_j.
$$

For ease of calculation we set $M = 1$ and remove the activation $\sigma$ to consider a single-head linear self attention (LSA) layer:

$$
\boldsymbol{h}_{N+1}^+ = \boldsymbol{h}_{N+1} + \frac{1}{N+1} \sum_{j=1}^{N+1} \langle Q \boldsymbol{h}_{N+1}, K \boldsymbol{h}_j \rangle V \boldsymbol{h}_j.
$$

Now we have

$$
[H_O]_{N+1}^+ - [H_P]_{N+1}^+
$$
$$
= \frac{\boldsymbol{h}_{N+1}^\top Q^\top K (\boldsymbol{h}_1^O V \boldsymbol{h}_1^O + \boldsymbol{h}_2^O V \boldsymbol{h}_2^O - \boldsymbol{h}_1^P V \boldsymbol{h}_1^P - \boldsymbol{h}_2^P V \boldsymbol{h}_2^P)}{N+1}.
$$

Denote

$$
R = Q^\top K = \begin{bmatrix} r_{11} & \cdots & r_{1D} \\ \vdots & & \vdots \\ r_{D1} & \cdots & r_{DD} \end{bmatrix} \in \mathbb{R}^{D \times D},
$$

$$
R_i(d) = \begin{bmatrix} r_{1i} \\ \vdots \\ r_{di} \end{bmatrix} \in \mathbb{R}^d. \tag{3}
$$

Then we have

$$
\boldsymbol{h}_{N+1}^\top Q^\top K \boldsymbol{h}_1^O
$$
$$
= \sum_{i=1}^{d} (\boldsymbol{x}_{N+1}^\top R_i(d) + r_{d+N+1,i}) \boldsymbol{x}_1^i
$$
$$
+ (\boldsymbol{x}_{N+1}^\top R_{d+1}(d) + r_{d+N+1,d+1}) y_1
$$
$$
+ (\boldsymbol{x}_{N+1}^\top R_{d+2}(d) + r_{d+N+1,d+2}),
$$

and similarly

$$
\boldsymbol{h}_{N+1}^\top Q^\top K \boldsymbol{h}_2^P
$$
$$
= \sum_{i=1}^{d} (\boldsymbol{x}_{N+1}^\top R_i(d) + r_{d+N+1,i}) \boldsymbol{x}_1^i
$$
$$
+ (\boldsymbol{x}_{N+1}^\top R_{d+1}(d) + r_{d+N+1,d+1}) y_1
$$
$$
+ (\boldsymbol{x}_{N+1}^\top R_{d+2}(d) + r_{d+N+1,d+3}).
$$

So we get

$$\boldsymbol{h}_{N+1}^\top Q^\top K \boldsymbol{h}_1^O - \boldsymbol{h}_{N+1}^\top Q^\top K \boldsymbol{h}_2^P$$

$$= r_{d+N+1,d+2} - r_{d+N+1,d+3}.$$

Similarly we can compute

$$V \boldsymbol{h}_1^O - V \boldsymbol{h}_2^P = \begin{bmatrix} v_{1,d+2} - v_{1,d+3} \\ \vdots \\ v_{D,d+2} - v_{D,d+3} \end{bmatrix}.$$

Notice that

$$\boldsymbol{h}_{N+1}^\top Q^\top K (\boldsymbol{h}_1^O V \boldsymbol{h}_1^O - \boldsymbol{h}_2^P V \boldsymbol{h}_2^P)$$
$$= \boldsymbol{h}_{N+1}^\top Q^\top K \boldsymbol{h}_2^P (V \boldsymbol{h}_1^O - V \boldsymbol{h}_2^P)$$
$$+ (\boldsymbol{h}_{N+1}^\top Q^\top K \boldsymbol{h}_1^O - \boldsymbol{h}_{N+1}^\top Q^\top K \boldsymbol{h}_2^P) V \boldsymbol{h}_2^P$$
$$+ (\boldsymbol{h}_{N+1}^\top Q^\top K \boldsymbol{h}_1^O - \boldsymbol{h}_{N+1}^\top Q^\top K \boldsymbol{h}_2^P)(V \boldsymbol{h}_1^O - V \boldsymbol{h}_2^P).$$

So the value change at the $d+1$-th row of the last column (where the output of the transformer should be stored) is $\boldsymbol{h}_{N+1}^\top Q^\top K \boldsymbol{h}_2^P (v_{d+1,d+2} - v_{d+1,d+3}) + v_{d+1} \boldsymbol{h}_2^P (r_{d+N+1,d+2} - r_{d+N+1,d+3}) + (v_{d+1,d+2} - v_{d+1,d+3})(r_{d+N+1,d+2} - r_{d+N+1,d+3})$, where $v_{d+1}$ is the $d+1$-th row of the matrix $V$.

Similarly we can compute the result for $\boldsymbol{h}_{N+1}^\top Q^\top K (\boldsymbol{h}_2^O V \boldsymbol{h}_2^O - \boldsymbol{h}_1^P V \boldsymbol{h}_1^P)$. If Assumption 3.1 stands, we have

$$|[\boldsymbol{h}_{N+1}^\top Q^\top K (\boldsymbol{h}_1^O V \boldsymbol{h}_1^O + \boldsymbol{h}_2^O V \boldsymbol{h}_2^O - \boldsymbol{h}_1^P V \boldsymbol{h}_1^P - \boldsymbol{h}_2^P V \boldsymbol{h}_2^P)]_{d+1}|$$
$$\leq |\boldsymbol{h}_{N+1}^\top Q^\top K (\boldsymbol{h}_1^P - \boldsymbol{h}_2^P)|\epsilon + |v_{d+1}(\boldsymbol{h}_1^P - \boldsymbol{h}_2^P)|\epsilon + 2\epsilon^2$$
$$\leq |\sum_{i=1}^{d} g_i(x_1^i - x_2^i) + g_{d+1}(y_1 - y_2)|\epsilon$$
$$+ |\sum_{i=1}^{d} v_{d+1,i}(\boldsymbol{x}_1^i - \boldsymbol{x}_2^i) + v_{d+1,d+1}(y_1 - y_2)|\epsilon + 2\epsilon^2$$
$$\leq \sqrt{|\sum_{i=1}^{d+1} g_i^2| |\sum_{i=1}^{d+1}(\boldsymbol{x}_1^i - \boldsymbol{x}_2^i)^2|}\epsilon$$
$$+ \sqrt{|\sum_{i=1}^{d+1} v_{d+1,i}^2| |\sum_{i=1}^{d+1}(\boldsymbol{x}_1^i - \boldsymbol{x}_2^i)^2|}\epsilon + 2\epsilon^2$$
$$= C(\boldsymbol{x}_{N+1}, Q, K, V)\sqrt{\sum_{i=1}^{d+1}(\boldsymbol{x}_1^i - \boldsymbol{x}_2^i)^2}\epsilon + 2\epsilon^2$$
$$= C(x_{N+1}, Q, K, V)X\epsilon + 2\epsilon^2,$$

where $g_i = \boldsymbol{x}_{N+1}^\top R_i(d) + r_{d+N+1,i}$, $C(\boldsymbol{x}_{N+1}, Q, K, V) = \sqrt{\sum_{i=1}^{d+1} g_i^2} + \sqrt{\sum_{i=1}^{d+1} v_{d+1,i}^2}$ and $X \sim \chi_{d+1}$ follows the chi distribution with $d+1$ degrees of freedom. From the induction it is clear that it doesn't matter whether the two permuted columns are adjacent or not. The above inequality always holds for a transposition and we only need to replace the $|\boldsymbol{h}_1^P - \boldsymbol{h}_2^P|$ term with $|\boldsymbol{h}_i^P - \boldsymbol{h}_j^P|$ for the transposition $(ij)$.

## 2. $k$ degree permutation

Now we consider a $k$ degree permutation of the prompt columns. According to group theory, each permutation can be written as a product of disjoint cycles, suppose there are a total of $P$ cycles and each cycle contains $a_p$ ($p = 1, \cdots, P$) elements, then apparently $\sum_{p=1}^{P} a_p = k$.

Moreover, each cycle can be written as a product of transpositions. For example, an $m$-cycle $(c_1 \cdots c_m) = (c_1 c_m) \cdots (c_1 c_3)(c_1 c_2)$. So every $m$-cycle can be written as a product of no more than $m$ transpositions, thus each $k$ degree permutation can be expressed as a product of no more than $\sum_{p=1}^{P} a_p = k$ transpositions. So from the above analysis we have

$$\mathbb{E}[|\hat{y}_{N+1} - y_{N+1}|]$$

$$\leq \frac{1}{N+1} \mathbb{E}[C(\boldsymbol{x}_{N+1}, Q, K, V) \sum_{j=1}^{k} X_d \epsilon + 2k\epsilon^2] \tag{4}$$

$$\rightarrow C(\boldsymbol{x}_{N+1}, Q, K, V) \frac{k\sqrt{d}}{N} \epsilon + \frac{2k}{N} \epsilon^2 \; (d, N \rightarrow \infty)$$

Here we used the fact that $\mathbb{E}[X_d] = \sqrt{2}\Gamma(\frac{1}{2}(d+2))/\Gamma(\frac{1}{2}(d+1))$. By Legendre duplication formula we rewrite the mean as

$$\mathbb{E}[X_d] = \sqrt{2/\pi} 2^{d-1} \frac{(\Gamma(d+1/2))^2}{\Gamma(d)}.$$

Now we use Stirling's approximation for Gamma function Define:

$$A = \sqrt{2\pi}((\frac{d+1}{2} - 1)^{\frac{d}{2}} e^{-(\frac{d-1}{2})} [1 + \frac{1}{12(\frac{d-1}{2})} + \mathcal{O}(\frac{1}{(d+1)^2})]),$$

$$B = \sqrt{2\pi}(d-1)^{d-\frac{1}{2}} e^{-(d-1)} [1 + \frac{1}{12(d-1)} + \mathcal{O}(\frac{1}{(d+1)^2})].$$

Then:

$$\mathbb{E}[X_d] = \sqrt{2/\pi} 2^{d-1} \cdot \frac{A^2}{B}$$

$$= (d-1)^{1/2} \cdot [1 + \frac{1}{4(d+1)} + \mathcal{O}(\frac{1}{(d+1)^2})]$$

$$= \sqrt{d}[1 - \frac{1}{4(d+1)} + \mathcal{O}(\frac{1}{(d+1)^2})],$$

thus we get the result for eq. (4). $\qquad \square$

**Corollary 3.1.** *There exist pretrained L-layer Transformers for which the difference bound in Theorem 3.1 remains valid, up to a factor of L.*

*Proof.* One can directly check that letting $r_{d+i,d+2} = r_{d+i,d+3}, i \in [N]$ and $v_{d+1,d+2} = v_{d+1,d+3}$ will ensure only the $d+1$-th row of the last column (where the prediction $\hat{y}_{N+1}$ should be stored) is changed when the input flows through a Transformer block (maintaining the position and value of of $(\boldsymbol{x}, y)$ in the input matrix), thus for every Transformer layer the error is at most

$$C_1 \frac{k\sqrt{d}}{N} \epsilon + \frac{2k}{N} \epsilon^2,$$

and the accumulative error should be bounded by $L$ times the above error. $\qquad \square$

**Theorem 3.2.** *Assume that each element of $\boldsymbol{x}_i$, denoted as $\boldsymbol{x}_i^k$, follows a normal distribution $\mathcal{N}(0, 1/2)$. For linear regression tasks $y_i = \boldsymbol{w}^\top \boldsymbol{x}_i$, let $\Delta y_{N+1} = \hat{y}_{N+1} - y_{N+1}$, where $\hat{y}_{N+1}$ represents a Transformer block's prediction using RoPE after applying a k-degree permutation to the prompt, and $y_{N+1}$ is the prediction based on the original prompt. Then, the following result holds:*

$$\sup \mathbb{E}[|\Delta y_{N+1}|] \longrightarrow C_{RoPE} \frac{kd^3}{N},$$

*where $C_{RoPE}$ is a constant that depends on $\boldsymbol{x}_{N+1}, Q, K, V, \Theta = \{\theta_i\}_i$.*

*Proof.* We first consider the case where only two adjacent columns are permuted and then generalize to random permutation between $k$ columns.

**1. Permutation between two adjacent columns with RoPE**

The original input with RoPE is:

$$H_O = \begin{bmatrix} \boldsymbol{x}_1 & \boldsymbol{x}_2 & \cdots & \boldsymbol{x}_N & \boldsymbol{x}_{N+1} \\ y_1 & y_2 & \cdots & y_N & 0 \end{bmatrix} \in \mathbb{R}^{(d+1)\times(N+1)}.$$

Note: With RoPE, positional information is encoded through the attention mechanism itself rather than explicit positional vectors. The rotary transformation is applied to queries and keys before computing attention scores.

After permutation between columns 1 and 2:

$$H_P = \begin{bmatrix} \boldsymbol{x}_2 & \boldsymbol{x}_1 & \cdots & \boldsymbol{x}_N & \boldsymbol{x}_{N+1} \\ y_2 & y_1 & \cdots & y_N & 0 \end{bmatrix} \in \mathbb{R}^{(d+1)\times(N+1)}.$$

The key difference with RoPE is in the attention computation. For a token embedding $\boldsymbol{h} = [\boldsymbol{x}, y]^\top$, the RoPE transformation for position $m$ is applied to the query and key projections:

For each dimension pair $(2i, 2i+1)$ in the query/key vectors, we apply the rotation:

$$\begin{bmatrix} q_{2i}^{(m)} \\ q_{2i+1}^{(m)} \end{bmatrix} = \begin{bmatrix} \cos m\theta_i & -\sin m\theta_i \\ \sin m\theta_i & \cos m\theta_i \end{bmatrix} \begin{bmatrix} q_{2i} \\ q_{2i+1} \end{bmatrix}$$

The attention score between position $m$ and $n$ becomes:

$$a_{m,n} = \frac{(\mathcal{R}_m Q \boldsymbol{h}_m)^\top (\mathcal{R}_n K \boldsymbol{h}_n)}{\sqrt{d_k}} = \frac{\boldsymbol{h}_m^\top Q^\top \mathcal{R}_{n-m} K \boldsymbol{h}_n}{\sqrt{d_k}}$$

where $\mathcal{R}_{n-m}$ is the block-diagonal rotation matrix for relative position $n-m$.

Now consider the output difference for the last column:

$$\boldsymbol{h}_{N+1}^+ = \boldsymbol{h}_{N+1} + \frac{1}{N+1} \sum_{j=1}^{N+1} \langle Q\boldsymbol{h}_{N+1}, \mathcal{R}_{j-(N+1)} K \boldsymbol{h}_j \rangle V \boldsymbol{h}_j.$$

The difference between original and permuted outputs is:

$$[H_O]_{N+1}^+ - [H_P]_{N+1}^+$$
$$= \frac{1}{N+1} \big( \langle Q\boldsymbol{h}_{N+1}, \mathcal{R}_{1-(N+1)} K \boldsymbol{h}_1 \rangle V \boldsymbol{h}_1 + \langle Q\boldsymbol{h}_{N+1}, \mathcal{R}_{2-(N+1)} K \boldsymbol{h}_2 \rangle V \boldsymbol{h}_2$$
$$- \langle Q\boldsymbol{h}_{N+1}, \mathcal{R}_{1-(N+1)} K \boldsymbol{h}_2 \rangle V \boldsymbol{h}_2 - \langle Q\boldsymbol{h}_{N+1}, \mathcal{R}_{2-(N+1)} K \boldsymbol{h}_1 \rangle V \boldsymbol{h}_1 \big)$$

Let $R_{-\Delta}^* = Q^\top \mathcal{R}_{-\Delta}^* K$. The core term becomes:

$$V\boldsymbol{h}_1 \boldsymbol{h}_{N+1}^\top (R_{-N}^* - R_{1-N}^*) \boldsymbol{h}_1 - V\boldsymbol{h}_2 \boldsymbol{h}_{N+1}^\top (R_{-N}^* - R_{1-N}^*) \boldsymbol{h}_2$$

The RoPE-specific effect appears in the difference $R_{-N}^* - R_{-N+1}^* = Q^\top (\mathcal{R}_{-N} - \mathcal{R}_{-N+1}) K$. For small rotation angles $\theta_i$, we can approximate:

$$\mathcal{R}_{-N} - \mathcal{R}_{-N+1} \approx \frac{d\mathcal{R}_{-\Delta}}{d\Delta} \bigg|_{\Delta=N}$$

Each 2×2 block of this derivative is:

$$\frac{d}{d\Delta} \begin{bmatrix} \cos \Delta\theta_i & -\sin \Delta\theta_i \\ \sin \Delta\theta_i & \cos \Delta\theta_i \end{bmatrix} = \theta_i \begin{bmatrix} -\sin \Delta\theta_i & -\cos \Delta\theta_i \\ \cos \Delta\theta_i & -\sin \Delta\theta_i \end{bmatrix}$$

Also we have

$$|\boldsymbol{h}_{N+1}^\top (R^*_{-N} - R^*_{1-N})\boldsymbol{h}_1|$$

$$= |\sum_{i=1}^d (\boldsymbol{x}_{N+1}^\top S_i(d) + s_{d+N+1,i})\boldsymbol{x}_1^i + (\boldsymbol{x}_{N+1}^\top S_{d+1}(d) + s_{d+N+1,d+1})y_1|$$

$$\leq a |\sum_{i=1}^d (1 + \boldsymbol{w}_i)\boldsymbol{x}_1^i|,$$

where $a$ is the uniform upper bound for $|\boldsymbol{x}_{N+1}^\top S_i(d) + s_{d+N+1,i}|$ in which $i = 1, \cdots, d+1$. Also $S = (R^*_{-N} - R^*_{1-N})$ and the definition of $S_i(d)$ and $s_{i,j}$ follows that in eq. (3). Note that $|s_{i,j}| \leq \|\boldsymbol{q}_i^\top\| \|\mathcal{R}_{-N} - \mathcal{R}_{1-N}\|_F \|\boldsymbol{k}_j\|$, so $a \leq b^2 d\sqrt{d} \cdot \max_i \theta_i$. Thus we have

$$|[V\boldsymbol{h}_1]_{d+1}\boldsymbol{h}_{N+1}^\top (R^*_{-N} - R^*_{1-N})\boldsymbol{h}_1|$$

$$\leq |[V\boldsymbol{h}_1]_{d+1}| |\boldsymbol{h}_{N+1}^\top (R^*_{-N} - R^*_{1-N})\boldsymbol{h}_1|$$

$$\leq va (\sum_{i=1}^d \boldsymbol{x}_1^i)^2$$

$$\leq vad (\sum_{i=1}^d (\boldsymbol{x}_1^i)^2)$$

$$\leq vb^2 d^{2.5} \theta_{\max} X_d,$$

where $v = \max\{v_{d+1,1}, \cdots, v_{d+1,d+1}\}$, $b = \max_i\{\|\boldsymbol{q}_i\|, \|\boldsymbol{k}_i\|\}$ and $X_d \sim \chi_d^2$.

**2. $k$ degree permutation with RoPE**

The same group-theoretic decomposition applies. Each $k$-degree permutation can be expressed as a product of at most $k$ transpositions. So the final bound becomes:

$$\mathbb{E}[|\hat{y}_{N+1} - y_{N+1}|]$$

$$\leq \frac{1}{N+1}\mathbb{E}\left[vb^2\theta_{\max}d^{2.5}\sum_{j=1}^k X_d\right] \tag{5}$$

$$\to C_{\text{RoPE}}(\boldsymbol{x}_{N+1}, Q, K, V, \Theta)\frac{kd^3}{N} \ (d, N \to \infty)$$

where $\Theta = \{\theta_i\}$ are the RoPE frequencies. $\qquad\square$

## C  PROOFS FOR SECTION 3.3

### C.1  USEFUL LEMMAS FOR IN-CONTEXT LEARNING

We first state the result for In-context Gradient Descent of the linear regression problem

$$L(\boldsymbol{w}) = \frac{1}{N}\sum_{j=1}^N (\boldsymbol{w}^\top \boldsymbol{x}_j - y_j)^2.$$

following Guo et al. (2023).

**Lemma C.1** (ICGD). *There exists an attention layer with 2 heads such that the following holds. For any input sequence $H$ that takes the form*

$$\boldsymbol{h}_i = [\boldsymbol{x}_i; y_i; \boldsymbol{w}; \boldsymbol{p}_i],$$

*the attention layer outputs*

$$\tilde{\boldsymbol{h}}_i = [\boldsymbol{x}_i; y_j; \tilde{\boldsymbol{w}}; \boldsymbol{p}_i],$$

*where $\tilde{\boldsymbol{w}}_i$ represents the result of one step of gradient descent*

$$\tilde{\boldsymbol{w}} = \boldsymbol{w} - \eta\nabla L(\boldsymbol{w}),$$

*for $i \in [N]$.*

*Proof.* We first define two attention heads $\{(Q_m, K_m, V_m)\}_{m=1,2}$ such that for all $i, j \in [N]$,

$$Q_1 \boldsymbol{h}_i = \begin{bmatrix} \boldsymbol{w} \\ -1 \\ \boldsymbol{0} \end{bmatrix}, K_1 \boldsymbol{h}_j = \begin{bmatrix} \boldsymbol{x}_j \\ y_j \\ \boldsymbol{0} \end{bmatrix}, V_1 \boldsymbol{h}_j = -\eta \begin{bmatrix} \boldsymbol{0}_{d+1} \\ \boldsymbol{x}_j \\ \boldsymbol{0} \end{bmatrix},$$

Thus for $i \in [N]$,

$$\langle Q_1 \boldsymbol{h}_i, K_1 \boldsymbol{h}_j \rangle - \langle Q_2 \boldsymbol{h}_i, K_2 \boldsymbol{h}_j \rangle = \boldsymbol{w}^\top \boldsymbol{x}_j - y_j$$

Therefore

$$\langle Q_1 \boldsymbol{h}_i, K_1 \boldsymbol{h}_j \rangle V_1 \boldsymbol{h}_j + \langle Q_2 \boldsymbol{h}_i, K_2 \boldsymbol{h}_j \rangle V_2 \boldsymbol{h}_j$$
$$= (\langle Q_1 \boldsymbol{h}_i, K_1 \boldsymbol{h}_j \rangle - \langle Q_2 \boldsymbol{h}_i, K_2 \boldsymbol{h}_j \rangle) \cdot \eta [\boldsymbol{0}_{d+1}; \boldsymbol{x}_j; \boldsymbol{0}]$$
$$= -\eta (\boldsymbol{w}^\top \boldsymbol{x}_j - y_j) \cdot [\boldsymbol{0}_{d+1}; \boldsymbol{x}_j; \boldsymbol{0}].$$

Summing the above for all $i \in [N]$ yields

$$\sum_{j=1}^N \sum_{m=1,2} \frac{1}{N} \langle Q_m \boldsymbol{h}_i, K_m \boldsymbol{h}_j \rangle V_m \boldsymbol{h}_j$$
$$= \frac{1}{N} [\sum_{j=1}^N -\eta (\boldsymbol{w}^\top \boldsymbol{x}_j - y_j)] \cdot [\boldsymbol{0}_{d+1}; \boldsymbol{x}_j; \boldsymbol{0}]$$
$$= [\boldsymbol{0}_{d+1}; -\eta \nabla L(\boldsymbol{w}); \boldsymbol{0}].$$

Thus the attention layer outputs

$$\tilde{\boldsymbol{h}}_i = \boldsymbol{h}_i + \sum_{m=1}^2 \sum_{j=1}^N \frac{1}{N} \langle Q_m \boldsymbol{h}_i, K_m \boldsymbol{h}_j \rangle V_m \boldsymbol{h}_j$$
$$= \begin{bmatrix} \boldsymbol{x}_i \\ y_i \\ \boldsymbol{w} \\ * \end{bmatrix} + \begin{bmatrix} \boldsymbol{0}_{d+1} \\ -\eta \nabla L(\boldsymbol{w}) \\ \boldsymbol{0} \end{bmatrix}$$
$$= \begin{bmatrix} \boldsymbol{x}_i \\ y_i \\ \boldsymbol{w} - \eta \nabla L(\boldsymbol{w}) \\ * \end{bmatrix}.$$

This finishes the proof. $\qquad\square$

**Lemma C.2** (In-context linear regression). *A Transformer with $\mathcal{O}(\epsilon^{-1})$ layers can implement in-context gradient descent such that its prediction $\hat{y}_i = [\text{TF}(H)]_{d+1,i}$ satisfies*

$$|\hat{y}_i - \langle \hat{\boldsymbol{w}}_i, \boldsymbol{x}_i \rangle| \le \epsilon.$$

The lemma directly follows Guo et al. (2023) Theorem B.5, so we omit the detailed proof and only provide two key steps. The first step is to determine the number of Transformer layers needed to achieve $\epsilon$ accuracy. The second step is to construct a linear prediction layer which stores the prediction of the Transformer (Guo et al. (2023) Lemma B.2).

**Lemma C.3.** *There exists an MLP layer with parameters $W_1, W_2$ such that $H' = \text{MLP}_{W_1, W_2}(H)$, where $H$ is the input with the one-hot positional encoding, and $H'$ is the input with the positional encoding $\bar{\boldsymbol{p}}_i = [\boldsymbol{0}_{N-3}; 1; i; i^2; i^3]$.*

*Proof.* We need to construct weight matrices not reliant on the input $h_i$ such that the one-hot PE $p_i$ can be transformed to the specific format in the Lemma and replace the original PE. Consider two matrices $P, Q$ which satisfies

$$P_{N+d-2:N+d+2,i} = [1; i; i^2; i^3], Q_{d+i+1,d+i+1} = -1$$

and other parts of $P, Q$ be 0. Recall that $h_i = [x_i; y_i; p_i; \mathbf{0}_{D-N-d-2}]$, then one can directly check that letting $W_2 = P + Q, W_1 = I$ yields

$$h_i' = h_i + W_2\sigma(W_1 h_i) = [x_i; y_i; \bar{p}_i; \mathbf{0}_{D-N-d-2}].$$

This shows that a single MLP layer can indeed change the input format in this specific way, and the weight matrices of the MLP layer doesn't rely on the input $x_i, y_i$, thus concluding the proof. $\qquad\square$

**Lemma C.4.** *There exists an MLP layer such that for the input $H$ of the form*

$$H = \begin{bmatrix} \boldsymbol{x}_1 & \boldsymbol{x}_2 & \cdots & \boldsymbol{x}_N & \boldsymbol{x}_{N+1} \\ \bar{\boldsymbol{p}}_1 & \bar{\boldsymbol{p}}_2 & \cdots & \bar{\boldsymbol{p}}_N & \bar{\boldsymbol{p}}_{N+1} \end{bmatrix},$$

*it outputs*

$$\mathrm{MLP}^{(1)}(\mathrm{H}) = \begin{bmatrix} \sigma_\rho(W_1 \boldsymbol{x}_1) & \cdots & \sigma_\rho(W_1 \boldsymbol{x}_{N+1}) \\ \boldsymbol{x}_1 & \cdots & \boldsymbol{x}_{N+1} \\ \bar{\boldsymbol{p}}_1' & \cdots & \bar{\boldsymbol{p}}_{N+1}' \end{bmatrix}.$$

*The $L + 1$ Transformer blocks that follows output*

$$\tilde{H} = \begin{bmatrix} \boldsymbol{x}_1 & \boldsymbol{x}_2 & \cdots & \boldsymbol{x}_N & \boldsymbol{x}_{N+1} \\ \mathbf{0}_d & \mathbf{0}_d & \cdots & \mathbf{0}_d & \mathbf{0}_d \\ \mathbf{0}_d & \boldsymbol{x}_1 & \cdots & \boldsymbol{x}_{N-1} & \boldsymbol{x}_N \\ \boldsymbol{x}_1 & \boldsymbol{x}_2 & \cdots & \boldsymbol{x}_N & \boldsymbol{x}_{N+1} \\ \tilde{\boldsymbol{p}}_1 & \tilde{\boldsymbol{p}}_2 & \cdots & \tilde{\boldsymbol{p}}_N & \tilde{\boldsymbol{p}}_{N+1} \end{bmatrix},$$

*where $\tilde{\boldsymbol{p}}_i, \boldsymbol{p}_i'$ differs from $\boldsymbol{p}_i$ only in the dimension of the zero paddings.*

*Proof.* For the first MLP layer, consider any input token $\boldsymbol{h}_i = [\boldsymbol{x}_i; \bar{\boldsymbol{p}}_i]$. Define weight matrices $W_1, W_2 \in \mathbb{R}^{D \times D}$ such that

$$W_1 \boldsymbol{h}_i = \begin{bmatrix} \pm\boldsymbol{x}_i \\ \pm\boldsymbol{x}_i \\ \pm\boldsymbol{x}_i \\ \mathbf{0} \end{bmatrix}, \sigma(W_1 \boldsymbol{h}_i) = \begin{bmatrix} \sigma(\pm\boldsymbol{x}_i) \\ \sigma(\pm\boldsymbol{x}_i) \\ \sigma(\pm\boldsymbol{x}_i) \\ \mathbf{0} \end{bmatrix},$$

$$W_2\sigma(W_1 \boldsymbol{h}_i) = \begin{bmatrix} \sigma(\boldsymbol{x}_i) - \sigma(-\boldsymbol{x}_i) \\ \mathbf{0} \end{bmatrix} + \begin{bmatrix} -\sigma(\boldsymbol{x}_i) + \sigma(-\boldsymbol{x}_i) \\ \mathbf{0} \end{bmatrix}$$

$$+ \begin{bmatrix} \mathbf{0}_d \\ \sigma(\boldsymbol{x}_i) - \sigma(-\boldsymbol{x}_i) \\ \mathbf{0} \end{bmatrix}.$$

Therefore, the output of the MLP layer is

$$\bar{\boldsymbol{h}}_i = \boldsymbol{h}_i + W_2\sigma(W_1 \boldsymbol{h}_i) = \begin{bmatrix} \boldsymbol{x}_i \\ \boldsymbol{x}_i \\ \bar{\boldsymbol{p}}_i \end{bmatrix}.$$

Now we need to achieve two things:

- Move the $\boldsymbol{x}_i$ into the $(3d + 1 : 4d)$ block in the final layer, which takes the same number of attention heads in every layer.

- Use one copying layer with a single attention head to copy each $\boldsymbol{x}_i$ to the $(2d + 1 : 3d)$ block of the next token.

$\qquad\square$

**Lemma 3.1.** *For any $\epsilon > 0$, there exists a Transformer with $\mathcal{O}(\epsilon^{-1})$ blocks such that for the input $\tilde{H}$ of the form*

$$\tilde{H} = \begin{bmatrix} \boldsymbol{x}_1 & \boldsymbol{x}_2 & \cdots & \boldsymbol{x}_N & \boldsymbol{x}_{N+1} \\ \mathbf{0}_d & \mathbf{0}_d & \cdots & \mathbf{0}_d & \mathbf{0}_d \\ \mathbf{0}_d & \boldsymbol{x}_1 & \cdots & \boldsymbol{x}_{N-1} & \boldsymbol{x}_N \\ \mathbf{0}_d & \boldsymbol{x}_2 & \cdots & \boldsymbol{x}_N & \boldsymbol{x}_{N+1} \\ \boldsymbol{p}_1 & \boldsymbol{p}_2 & \cdots & \boldsymbol{p}_N & \boldsymbol{p}_{N+1} \end{bmatrix},$$

*the prediction of the Transformer $\hat{\boldsymbol{y}}_i = [\mathrm{TF}(\tilde{H})]_{(d+1):2d,i} (i \in [N+1])$ satisfies*

$$\|\hat{\boldsymbol{y}}_i - A\boldsymbol{x}_i\|_2 \leq \sqrt{d}\epsilon,$$

*with $d$ being the dimension of $\boldsymbol{x}$.*

*Proof.* For the dynamical system we have the loss function

$$\hat{L}(A) = \frac{1}{N} \sum_{j=1}^{N} \|A\boldsymbol{x}_j - \boldsymbol{y}_j\|_2^2,$$

where $\boldsymbol{y}_j = \boldsymbol{x}_{j+1}$. The multi-output dynamic system problem is equivalent to $d$ separable single-output linear regression problems, one for each output dimension. So the proof follows by directly repeating the analysis in Lemma C.2, with the following adaptation

- Use a transformer with $2d$ heads to perform $d$ parallel linear regression problems (each with 2 heads), using in-context gradient descent (Lemma C.1) as the internal optimization algorithm.

- Use a single-attention layer with $d$ parallel linear prediction heads to writ prediction $(\hat{\boldsymbol{y}}_i)_j$ into location $(i, d+j)$ with $|(\hat{\boldsymbol{y}}_i)_j - \left\langle (\hat{A}_i)_j, \boldsymbol{x}_i \right\rangle| \leq \epsilon$.

$\square$

## C.2 PROOFS FOR MAIN THEOREMS

**Theorem 3.3.** *Under Assumption 3.1, assume $\boldsymbol{x}_0^k \sim \mathcal{N}(0, 1/2)$. For first order difference equation $\boldsymbol{x}_{i+1} = A\boldsymbol{x}_i + \boldsymbol{b}$, define $\Delta\boldsymbol{x}_{N+1} = \hat{\boldsymbol{x}}_{N+1} - \boldsymbol{x}_{N+1}$, where $\hat{\boldsymbol{x}}_{N+1}$ represents the transformer's prediction after applying a $k$-degree permutation to the prompt, and $\boldsymbol{x}_{N+1}$ corresponds to the prediction based on the original prompt. Then the following result holds:*

$$\sup \mathbb{E}[\|\Delta\boldsymbol{x}_{N+1}\|_2] \longrightarrow C_2 \frac{kd}{N}\epsilon + \frac{2k\sqrt{d}}{N}\epsilon^2 \ (d, N \to \infty),$$

*where $C_2$ is a constant dependent on $\boldsymbol{x}_{N+1}, Q, K, V$.*

*Proof.* We inherit the proof in Theorem 3.1 by setting $y_i = 0$. W.L.O.G. we assume $\|A\| = 1$ and $b = 0$. Recall the input matrix

$$H = \begin{bmatrix} \boldsymbol{x}_1 & \boldsymbol{x}_2 & \cdots & \boldsymbol{x}_N & 0 \\ 0 & 0 & \cdots & 0 & 0 \\ \boldsymbol{p}_1 & \boldsymbol{p}_2 & \cdots & \boldsymbol{p}_N & \boldsymbol{p}_{N+1} \\ \boldsymbol{0} & \boldsymbol{0} & \cdots & \boldsymbol{0} & \boldsymbol{0} \end{bmatrix} \in \mathbb{R}^{D \times (N+1)},$$

so when we swap column 1 and column 2 we still have

$$[H_O]_{N+1}^+ - [H_P]_{N+1}^+$$
$$= \frac{h_{N+1}^\top Q^\top K (h_1^O V h_1^O + h_2^O V h_2^O - h_1^P V h_1^P - h_2^P V h_2^P)}{N+1}.$$

Replacing $x_{N+1}$ and $y_i, i \in [N+1]$ in the proof of Theorem 1 with 0 yields the value change at the $j$-th row ($j \in [d]$) of the last column is:

$$|[\boldsymbol{h}_{N+1}^\top Q^\top K(\boldsymbol{h}_1^O V \boldsymbol{h}_1^O + \boldsymbol{h}_2^O V \boldsymbol{h}_2^O - \boldsymbol{h}_1^P V \boldsymbol{h}_1^P - \boldsymbol{h}_2^P V \boldsymbol{h}_2^P)]_j|$$

$$\leq |\boldsymbol{h}_{N+1}^\top Q^\top K(\boldsymbol{h}_1^P - \boldsymbol{h}_2^P)|\epsilon + |v_j(\boldsymbol{h}_1^P - \boldsymbol{h}_2^P)|\epsilon + 2\epsilon^2$$

$$\leq |\sum_{i=1}^d g_i(\boldsymbol{x}_1^i - \boldsymbol{x}_2^i)|\epsilon + |\sum_{i=1}^d v_{j,i}(\boldsymbol{x}_1^i - \boldsymbol{x}_2^i)|\epsilon + 2\epsilon^2$$

$$\leq \sqrt{|\sum_{i=1}^d g_i^2||\sum_{i=1}^d (\boldsymbol{x}_1^i - \boldsymbol{x}_2^i)^2|}\epsilon$$

$$+ \sqrt{|\sum_{i=1}^d v_{j,i}^2||\sum_{i=1}^d (\boldsymbol{x}_1^i - \boldsymbol{x}_2^i)^2|}\epsilon + 2\epsilon^2$$

$$= C_2 X\epsilon + 2\epsilon^2,$$

where $C_2 = \sqrt{\sum_{i=1}^d g_i^2} + \sqrt{\sum_{i=1}^d v_{j,i}^2}$. $X = \sqrt{\sum_{i=1}^d (x_1^i - x_2^i)^2}$. Note that $x_1^i = \sum_{j=1}^d a_{ij} x_0^j \sim \mathcal{N}(0, (\sum_j a_{ij}^2))$. Now suppose $\sum_j a_{ij}^2 = 1$ for $i \in [d]$, then we still have $X \sim \chi_{d+1}$, and the rest is the same as the proof in Theorem 3.1, except that the $L_2$ norm should be multiplied by $\sqrt{d}$ since the prediction is a $d$-dimension vector instead of a number. $\qquad\square$

**Theorem 3.4.** *For any $\epsilon > 0$, there exists a Transformer with $\mathcal{O}(\epsilon^{-1})$ layers such that for an input structured as described in eq. (2), it implements approximate GD on the input with shifted prompt order and the prediction for $\boldsymbol{x}_i$ ($i \in [N]$) satisfies the following upper bound:*

$$\|\hat{\boldsymbol{x}}_{i+1} - A\boldsymbol{x}_i\|_2 \leq (\sqrt{k}d + \sqrt{d})\epsilon,$$

*where $k, d$ represents the degree of permutation and the dimension of $\boldsymbol{x}$ respectively.*

*Proof.* We first consider the simple case of flipping the first two tokens of the input, resulting in the input format

$$H = \begin{bmatrix} \boldsymbol{x}_2 & \boldsymbol{x}_1 & \cdots & \boldsymbol{x}_N & \boldsymbol{x}_{N+1} \\ \bar{\boldsymbol{p}}_1 & \bar{\boldsymbol{p}}_2 & \cdots & \bar{\boldsymbol{p}}_N & \bar{\boldsymbol{p}}_{N+1\cdot} \end{bmatrix}$$

Following the matrix transformation procedure in Lemmas 3.1 and C.4, we get the input format in Lemma 3.1

$$\tilde{H} = \begin{bmatrix} \boldsymbol{x}_2 & \boldsymbol{x}_1 & \boldsymbol{x}_3 & \cdots & \boldsymbol{x}_{N+1} \\ \boldsymbol{0}_d & \boldsymbol{0}_d & \boldsymbol{0}_d & \cdots & \boldsymbol{0}_d \\ \boldsymbol{0}_d & \boldsymbol{x}_2 & \boldsymbol{x}_1 & \cdots & \boldsymbol{x}_N \\ \boldsymbol{0}_d & \boldsymbol{x}_1 & \boldsymbol{x}_3 & \cdots & \boldsymbol{x}_{N+1} \\ \boldsymbol{p}_1 & \boldsymbol{p}_2 & \boldsymbol{p}_3 & \cdots & \boldsymbol{p}_{N+1} \end{bmatrix},$$

where the prediction corresponding to $\boldsymbol{x}_1, \boldsymbol{x}_2$ is changed from $\boldsymbol{x}_2, \boldsymbol{x}_3$ to $\boldsymbol{x}_3, \boldsymbol{x}_1$ respectively. Notice that the Transformer implements in-context gradient descent by Lemma C.1, the gradient for the first element of the objective vector is

$$\nabla L(\boldsymbol{w}) = \frac{1}{N}\sum_{j=1}^N (\boldsymbol{w}^\top \boldsymbol{x}_j - y_j)\boldsymbol{x}_j.$$

Here $y_j = \boldsymbol{x}_{j+1}^1$. But for the permuted input, the gradient becomes

$$\nabla L'(\boldsymbol{w}) = \frac{1}{N}\sum_{j=1}^N (\boldsymbol{w}^\top \boldsymbol{x}_j - y_j')\boldsymbol{x}_j,$$

where $y_1' = \boldsymbol{x}_3^1, y_2' = \boldsymbol{x}_1^1$, and $y_j' = y_j$ for $j \geq 3$. So the difference in gradient is

$$\boldsymbol{e} = \frac{\boldsymbol{x}_3^1 - \boldsymbol{x}_2^1}{N}\boldsymbol{x}_1 + \frac{\boldsymbol{x}_1^1 - \boldsymbol{x}_3^1}{N}\boldsymbol{x}_2.$$

So the gradient descent update at each iteration $t$ is

$$\boldsymbol{w}_{t+1} = \boldsymbol{w}_t - \eta \nabla L(\boldsymbol{w}) - \eta \boldsymbol{e},$$

and after $T$ rounds the parameter $\boldsymbol{w}$ becomes

$$\boldsymbol{w}_T = \boldsymbol{w}_0 - \eta \sum_{t=0}^{T-1} \nabla L(\boldsymbol{w}_t) - \eta T \boldsymbol{e}.$$

Thus the cumulative error term induced by $\boldsymbol{e}$ is

$$\eta T \boldsymbol{e} = 2\eta T \cdot \frac{u_1 - u_2}{N} \boldsymbol{x},$$

where $u_i \sim \mathcal{N}(0, 1/2)$, $i = 1, 2$ and each component of $\boldsymbol{x}$ also follows $\mathcal{N}(0, 1/2)$. Thus the expectation of the squared error is

$$\mathbb{E}[\|\eta T \boldsymbol{e}\|^2] = (\frac{2\eta T}{N})^2 \mathbb{E}[\|u_1 - u_2\|^2] \mathbb{E}[\|\boldsymbol{x}\|^2]$$
$$= \frac{2d\eta^2 T^2}{N^2}.$$

Here we used the fact that $\mathbb{E}[\|u_1 - u_2\|^2] = 1$ and $\mathbb{E}[\|x\|^2] = d/2$. Note that $T$ is the number of layer of the Transformer as one layer of Transformer implements one step of gradient descent. Lemma C.2 states that to achieve $\mathcal{O}(\epsilon)$ accuracy we need a Transformer with $\mathcal{O}(\epsilon^{-1})$ layers, thus choosing $\eta = \mathcal{O}(\epsilon)$ would yield

$$\sqrt{\mathbb{E}[\|\eta T \boldsymbol{e}\|^2]} \le \frac{\sqrt{d}}{N}.$$

Since $\boldsymbol{x}$ is bounded, the final $L_2$ norm of the error brought by the approximate gradient descent should be bounded by

$$\sqrt{d}(\frac{\sqrt{d}}{N} + \epsilon) = \frac{d}{N} + \sqrt{d}\epsilon.$$

By induction a $k$-degree permutation on the prompt input would yield a final error of

$$\frac{\sqrt{k}d}{N} + \sqrt{d}\epsilon,$$

and choosing $N = \mathcal{O}(\epsilon^{-1})$ would yield the desired result. $\qquad\square$

