# OpenReview forum: "On the Effect of Positional Encoding for In-context Learning in Transformers"
_ICLR.cc/2026/Conference — Submitted to ICLR 2026_

### Official Review · Reviewer_m21Y · 2025-10-22

**Soundness:** 3
**Presentation:** 3
**Contribution:** 3
**Rating:** 6
**Confidence:** 2

**Summary:**

This paper provides a theoretical analysis of how positional encoding (PE) affects the in-context learning (ICL) capability of transformers. It derives sufficient conditions on the transformer’s weight matrices that guarantee permutation invariance. For both linear regression and dynamical system tasks, it also proves that the error scales as $O(k/N)$ when $N$ is sufficiently large, demonstrating robustness of Transformers to prompt perturbations.

**Strengths:**

This paper derives sufficient conditions on the weight matrices that ensure permutation invariance in transformers with PE. Moreover, it quantitatively provides error bounds of  $O(k/N)$ for both linear regression and dynamical system task. Overall, this paper makes a valuable contribution, showing that Transformers remain robust to the permutation.

**Weaknesses:**

While this paper is theoretically strong and sufficiently novel, it is currently limited to linear tasks and considers only absolute positional encoding.
However, I think that these limitations are acceptable for the scope of this paper. Out of interest, I would like to ask the questions below.

**Questions:**

1. In the nonlinear case (e.g., logistic regression), would a similar quantitative analysis of output error bound be possible ? Although the error bound may differ from $O(k/N)$, could we still expect the bound decreasing when $N$ is large?

2. For other types of PE (e.g., Relative PE, Rotary PE), do you expect similar theoretical interpretations to hold, or would the underlying analysis have to be fundamentally different?

---

> ### Author Response · Authors · 2025-11-21
>
> Thank you for your thoughtful review of our paper and for providing these insightful comments. We address each point in our responses below.
>
>
> ### Question 1
> For nonlinear case the quantitative analysis is sill possible. for logistic regression (simplified) we need to predict $y = e^{w^\top x}$. Putting this into our analysis we need to compute $\mathbb{E}[e^{w^\top x}]$, this is of course possible if we impose certain assumptions on the data distribution of $x$, and this term would only be reflected in the numerator of the final order, so the bound would still be decreasing when $N$ is large.
>
> ### Question 2
> Regarding RoPE, we've added section 3.2.2 to explore its effect on ICL performance and showed that the $O(k/N)$ order still preserves but the reliance on $d$ grows. Please refer to official comment 'Summary of our paper revision' for more details.

---

> > ### Comment · Reviewer_m21Y · 2025-11-26
> >
> > Thank you very much for the clarification. I would like to keep my score.

---

### Official Review · Reviewer_GKie · 2025-10-31

**Soundness:** 2
**Presentation:** 2
**Contribution:** 2
**Rating:** 4
**Confidence:** 3

**Summary:**

This is my second review of this manuscript. Compared to the previous version, there are no substantial changes in this revision.

This paper mainly investigates how positional encoding and different permutations of in-context examples affect the model’s output in in-context learning tasks. The authors provide theoretical results showing that positional encoding can enhance robustness to perturbations in the prompt order, and they further validate these findings with experiments on linear regression and dynamical system tasks.

**Strengths:**

1. The paper provides a mathematical quantification of the relationship between positional encoding and input order perturbations, which is novel.

2. It designs two empirical tasks — one order-dependent and one order-independent — to validate the theory, demonstrating a well-reasoned experimental setup.

**Weaknesses:**

1. Multiple experimental runs were not conducted, and the chosen tasks are relatively simple.

2. The positional encoding used in this paper appears to impose certain constraints on the input length of the model.

**Questions:**

1. I noticed that in the equation on page 4, line 202, the authors omitted the higher-order terms of $B$. In this case, is it still rigorous to use the equality sign?
2. I have some questions regarding the one-hot positional encoding used by the authors, and please correct me if I have misunderstood. If the hidden state dimension $D$ is fixed, since the positional encoding satisfies $p_i \in \mathbb{R}^N$, this implies that the number or length of in-context examples cannot exceed $D$ — more precisely, it cannot exceed $D - d - 2$. In other words, the number of in-context examples (or context length) is constrained by the dimension of the hidden state, which should generally not be the case. We would consider raising the score if the authors could provide further clarification on this issue.
3. I suggest that the authors perform multiple experiments on the same synthetic dataset with different model initializations to improve the robustness of the results. It would also strengthen the paper if more complex (preferably nonlinear) models and tasks were included in the experiments to enhance persuasiveness and generality.

---

> ### Author Response · Authors · 2025-11-21
>
> Thank you for your thoughtful review of our paper and for providing these insightful comments. We address each point in our responses below.
>
>
> ### Weakness 1
>
> Each data point in our experiments is averaged over 10,000 runs to approximate the expectation $\mathbb{E}[\Delta y]$ in our theoretical results, as mentioned in section 4 "Linear regression" and "First Order Difference Equation" parts. The tasks are chosen to align with our theorems.
>
> ### Weakness 2 and Question 2
>
> We'd like to respond to this issue in three ways. First, the one-hot positional encoding format concatenated with the input matrix has been established as a convention in previous theoretical works [1-3]. Second, while the hidden state dimension is fixed, there's also a max\_prompt\_length or max\_seq\_length parameter when using the model.generate() api. These two parameters usually does not exceed 4096 (or 8192) tokens for small-scale GPT2 models considered in our setting. So there're really not so many in-context examples as each example may contain hundreds of tokens. Moreover, in real life a common setting is few-shot ICL, i.e. four-shot for evaluation on MATH500 in the huggingface repo, so actually $D > n$ is the common case. Third, we've added section 3.2.2 to consider the RoPE instead of one-hot PE to avoid the dependence of ICL examples on $D$, for details please see official comment 'Summary of our paper revision'.
>
> ### Question 1
>
> Thank you for raising this point, we have changed the $=$ to $\approx$.
>
> ### Question 3
>
> Thank you for the suggestion. While we acknowledge the value of testing different model initializations, we expect this would yield little variation in our results, as transformer models typically demonstrate robustness to initialization, particularly on simple tasks. Additionally, such an approach would be computationally inefficient, as it would require pretraining multiple models for each task. However, we agree that expanding our experimental analysis to include nonlinear models is valuable. Although our current experiments focus on linear self attention to maintain alignment with our theoretical framework, we plan to include experiments on nonlinear activation transformers in the appendix of a future version.
>
>
> [1] Bai, Y., Chen, F., Wang, H., Xiong, C., & Mei, S. (2023). Transformers as statisticians: Provable in-context learning with in-context algorithm selection. Advances in neural information processing systems, 36, 57125-57211.
>
> [2] Guo, T., Hu, W., Mei, S., Wang, H., Xiong, C., Savarese, S., & Bai, Y. (2023). How do transformers learn in-context beyond simple functions? a case study on learning with representations. arXiv preprint arXiv:2310.10616.
>
> [3] Wang, Z., Jiang, B., & Li, S. (2024, July). In-context learning on function classes unveiled for transformers. In Forty-first International Conference on Machine Learning.

---

### Official Review · Reviewer_yX6J · 2025-11-01

**Soundness:** 2
**Presentation:** 2
**Contribution:** 1
**Rating:** 2
**Confidence:** 5

**Summary:**

This paper theoretically investigates how positional encoding (PE) affects in-context learning in Transformers when the ordering in the prompt changes. The authors use one-hot positional encoding and prove that prediction changes as O(k/N) where k is the permutation degree and N is the number of examples, for both linear regression and first order difference equations (differential?). They show that under specific weight matrix conditions, Transformers can maintain permutation invariance despite PE. Experiments on synthetic tasks validate the O(k/N) scaling relationship. The authors claim PE enables robust ICL on order-sensitive tasks.

**Strengths:**

S1: The paper provides clean theoretical bounds (Theorems 3.1, 3.2) with explicit convergence rates for changes in the prompt order dependence.

**Weaknesses:**

W1: The impact of the core contribution is unclear. The fast that positional encoding affects output when prompt order changes is quite an expected result. The "mystery" that transformers show order sensitivity despite architectural permutation invariance is immediately resolved by noting PE exists.

W2: Mostly theoretical contributions. It is hard to extend these theoretical contributions to actionable insights. In fact, it isn't even clear if these theorems help growing intuition for how ICL should behave in real language models. Moreover, the theory doesn't seem to predict new phenomena.

W2: One-hot PE is completely disconnected from reality. As far as I know, no modern LLMs, or even transformers trained for specific purposes. uses one-hot positional encoding. How these results would impact RoPE embeddings should have been discussed.

W3: Existence proofs are hardly surprising. Proposition 1 and Condition 3.1 show that "there exist" Transformers satisfying certain properties. But it is hard to really understand the impact of such statement. In reality, a vast amount of computation can be expressed from a Transformer, and the existence of such a weight isn't surprising, nor belief changing.

W4: The "dynamical systems" task is ill presented or explained.

W5: There are no validation or even qualitative checks on real language models or other transformers trained on real data.

**Questions:**

Q1: How do these results extend to RoPE or any PE scheme actually used in modern LLMs?
Q2: Are there any results on real-data trained Transformers which even qualitatively justify the main finding?

---

> ### Author Response · Authors · 2025-11-21
>
> Thank you for your thoughtful review of our paper and for providing these insightful comments. We address each point in our responses below.
>
> ### Weakness 1 and 2
>
> Our major conclusion is not PE affects output but rather how it affects output. The intuition behind this paper is based on an empirical work [1] where they found that prompt order affects ICL performance drastically, and their solution is to find "fantastic order" of prompts. What our paper is suggesting is that perhaps we don't need to identify the "correct order" of prompts, we can simply add more examples to achieve robust ICL performance.
>
> ### Weakness 2' (there're two W2's so this is the second W2) and Question 1
>
> We have added a theorem and the corresponding experiment for RoPE, please see official comment 'Summary of our paper revision'.
>
> ### Weakness 3
>
> Condition 3.1 and Proposition 1 are presented primarily to offer high-level insight and to suggest the plausibility that a pretrained model satisfying such properties may exist. These elements are not central to our main theoretical contributions. We also emphasize that Proposition 1 indicates that under Condition 3.1, positional encodings may have no effect on in-context learning performance -- a finding that contradicts the "expected result" mentioned in Weakness 1.
>
> ### Weakness 4
>
> Could you please clarify further on this point?
>
> ### Weakness 5 and Question 2
>
> Our aim is not to explain the behavior of LLMs, but to rigorously analyze the transformer architecture itself. The complexity of real-world data often poses challenges for mathematical formalization, which is a primary reason why theoretical investigations of in-context learning (ICL) predominantly focus on linear regression tasks and related variants [2-5].
>
>
> [1] Lu, Y., Bartolo, M., Moore, A., Riedel, S., & Stenetorp, P. (2022, May). Fantastically ordered prompts and where to find them: Overcoming few-shot prompt order sensitivity. In Proceedings of the 60th Annual Meeting of the Association for Computational Linguistics (Volume 1: Long Papers) (pp. 8086-8098).
>
> [2] Von Oswald, J., Niklasson, E., Randazzo, E., Sacramento, J., Mordvintsev, A., Zhmoginov, A., & Vladymyrov, M. (2023, July). Transformers learn in-context by gradient descent. In International Conference on Machine Learning (pp. 35151-35174). PMLR.
>
> [3] Ahn, K., Cheng, X., Daneshmand, H., & Sra, S. (2023). Transformers learn to implement preconditioned gradient descent for in-context learning. Advances in Neural Information Processing Systems, 36, 45614-45650.
>
> [4] Gatmiry, K., Saunshi, N., Reddi, S. J., Jegelka, S., & Kumar, S. (2024). Can looped transformers learn to implement multi-step gradient descent for in-context learning?. arXiv preprint arXiv:2410.08292.
>
> [5] Bai, Y., Chen, F., Wang, H., Xiong, C., & Mei, S. (2023). Transformers as statisticians: Provable in-context learning with in-context algorithm selection. Advances in neural information processing systems, 36, 57125-57211.

---

> > ### Comment · Reviewer_yX6J · 2025-11-28
> >
> > I thank the author for their careful replies!
> >
> > ---
> >
> > W1: The statement that "we can simply add more examples to achieve robust ICL performance" seems quite orthogonal to the fantastic order claim: in real life you often don't have more examples, or hit a context length limit, etc, this doesn't seems to be a fair comparison, nor an unexpected one.
> >
> > W2 (the first W2, sorry about the misformatting): The mostly theoretical contributions remain a major weakness. For example proposition 1, that there exists such an permutation invariant ICL transformer isn't surprising, nor actionable.
> >
> > W2 (the second W2): Thank you this is a nice addition.
> >
> > W3: Does condition 3.1 hold (or expected to hold) in realistic settings? i.e. LLMs? If not, these lines or arguments further emphasize the weakness of the paper.
> >
> > W4: The first order differential equation tasks could be better illustrated with intuitive figures.
> >
> > ---
> >
> > Overall, I maintain my original assessment. While this paper provides theoretical contributions, its findings are not well ecologically grounded to practical scenarios and the presentation is not friendly to practitioners. A proper experimental grounding of why these theoretical findings are belief changing or providing better intuition for practical purposes would be crucial. Again, I understand  theoretical investigations are only amenable in very simplified settings, however a good theory paper should present a nuanced view on how well their assumptions hold, what are their practically relevant and falsifiable claims, and present the main findings in a more approachable way.

---

### Official Review · Reviewer_K6Rr · 2025-11-01

**Soundness:** 3
**Presentation:** 3
**Contribution:** 2
**Rating:** 6
**Confidence:** 4

**Summary:**

This paper theoretically studies how positional encoding (PE) affects the in-context learning (ICL) capabilities of Transformers. The authors consider two representative tasks—linear regression (order-invariant) and first-order dynamical systems (order-sensitive)—and establish formal results showing that (1) PE introduces bounded deviations in output under prompt permutation, with errors scaling as O(k/N); and (2) for dynamical systems, PE enables approximate gradient descent behavior, providing robustness to order changes. Their experiments confirm the theoretical claims.

**Strengths:**

1. This paper studys an important yet underexplored aspect of ICL—how prompt order and positional encoding interact and provide interesting theoretical results.
2. This paper is clearly written with well-motivated sectioning.

**Weaknesses:**

1. The main theoretical results are mostly constructive in nature (e.g., there exists a Transformer). However, it remains questionable whether real-world Transformers actually behave according to these theoretically constructed results.

2. The theoretical results are not particularly surprising — constructing specific Transformers to demonstrate how positional encoding improves performance on time-series tasks is rather expected.

3. Experiments focus on synthetic data. Adding natural sequential tasks or ablations with alternative PE types (e.g., RoPE, ALiBi) would strengthen the contribution.

**Questions:**

1. Could the authors provide more experiments on natural sequential tasks or include ablations with alternative positional encoding types (e.g., RoPE, ALiBi)?

2. Could the authors provide some theoretical results that go beyond the constructive setting and better reflect realistic scenarios?

---

> ### Author Response · Authors · 2025-11-21
>
> Thank you for your thoughtful review of our paper and for providing these insightful comments. We address each point in our responses below.
>
>
> ### Weakness 1
> Our main results Theorem 3.1, 3.2 and 3.3 (in revised pdf) does not rely on specific constructions but only on assumption 3.1. Assumption 3.1 only requires that two elements of  $Q^\top K$  and  V  be sufficiently close, which is a notably loose constraint. Furthermore, replacing this assumption with a general norm bound on the weight matrices would preserve our main conclusion regarding the $ \mathcal{O}(k/N) $ error rate. We introduce Assumption 3.1 primarily to ensure the perturbation  $\Delta y_{N+1}$  remains minimal even for small values of  $N$.
>
> ### Weakness 2
> PE improves ICL in time-series task is not the only result of our paper. The intuition behind this paper is based on an empirical work [1] where they found that prompt order affects ICL performance drastically, and their solution is to find "fantastic order" of prompts. Theorem 3.1-3.3 are actually suggesting that perhaps we don't need to identify the "correct order" of prompts, we can simply add more examples to achieve robust ICL performance.
>
> ### Weakness 3 and Question 1
> We've added experiments on RoPE in the revised pdf (please see official comment 'Summary of our paper revision').
>
>
> ### Question 2
> We've added results regarding RoPE (theorem 3.2) and the corresponding experiments. Also as mentioned in the rebuttal to weakness 1 our main results are not constructive.
>
> [1] Lu, Y., Bartolo, M., Moore, A., Riedel, S., & Stenetorp, P. (2022, May). Fantastically ordered prompts and where to find them: Overcoming few-shot prompt order sensitivity. In Proceedings of the 60th Annual Meeting of the Association for Computational Linguistics (Volume 1: Long Papers) (pp. 8086-8098).

---

### Official Review · Reviewer_DDwu · 2025-11-01

**Soundness:** 3
**Presentation:** 3
**Contribution:** 2
**Rating:** 4
**Confidence:** 3

**Summary:**

This paper analyzes how positional encodings (PEs) shape in-context learning (ICL) when the order of exemplars is perturbed: under an idealized attention model with explicit absolute positions, it proves output sensitivity to a (k)-degree permutation shrinks roughly as (O(k/N)) with the number of shots (N), gives a sufficient condition under which attention remains effectively permutation-invariant, and provides a constructive mechanism showing that PEs can implement approximate gradient-descent–style updates on simple sequential tasks; the trends are validated on synthetic linear regression and first-order dynamical systems, and the results motivate practical prompt/evaluation hygiene (fix or average exemplar order, use more shots, small permutation ensembles), though external validity for modern softmax attention and popular PEs (RoPE/ALiBi/learned APE) is not established.

**Strengths:**

*   A novel analysis that formalizes bounds linking order sensitivity in ICL to permutation degree (*k*) and the number of shots (*N*).
*   A constructive mechanism (approximate gradient descent) that links explicit position signals to robustness on order-sensitive sequence tasks.
*   Clean synthetic experiments whose trends match the theory, yielding actionable recommendations for prompt and evaluation hygiene (e.g., fixing/averaging order, adding shots, light ensembling over orders).
*   The paper is well-presented, clearly articulating when positions break permutation invariance in ways that are beneficial for ICL behavior.

**Weaknesses:**

*   The analysis targets an idealized setting (concatenated one-hot absolute PE + linearized attention), so its applicability to the more common transformer setting of softmax attention with RoPE, ALiBi, or learned PEs remains unclear.
*   Generality to complex, real-world data is unproven, with no tests on pretrained LLMs or on heavy-tailed/multimodal inputs.
*   The sufficient conditions (e.g., for permutation-invariance; matrix closeness assumptions) seem strong and are not shown to emerge during standard pretraining.
*   Experiments are small-scale and synthetic; there is no ablation on modern architectures or reporting of the constants/prefactors that would govern real-world effect sizes.

**Questions:**

*   Do the *O(k/N)* trends and robustness claims hold for non-Gaussian, heavy-tailed, or multimodal real-world datasets, and can they be observed in small open-source LLMs with standard PEs?
*   Can the theory (or new bounds) be extended to sinusoidal/learned APE, RoPE, and ALiBi under softmax attention? If not, where exactly does the analysis break, preventing it from guiding our understanding of real ICL?
*   Would it be possible to empirically measure how closely pretrained models satisfy your sufficient conditions? Could these conditions be relaxed or reinterpreted to make them more verifiable in practice?

---

> ### Author Response · Authors · 2025-11-21
>
> Thank you for your thoughtful review of our paper and for providing these insightful comments. We address each point in our responses below.
>
> ### Weakness 1
> The analysis setting indeed adopts some simplification, and we'd like to note that linear self attention is the standard setting in a line of theoretical works [1-4] as well as one-hot PE [5-7]. Also, we extended our one-hot PE results to RoPE and have updated our paper (please refer to official comment 'Summary of our paper revision').
>
> ### Weaknesses 2 and 4
> The complexity of real-world data often poses challenges for mathematical formalization, which is a primary reason why theoretical investigations of in-context learning (ICL) predominantly focus on linear regression tasks and related variants. Furthermore, our aim is not to explain the behavior of LLMs, but to rigorously analyze the transformer architecture itself.
>
> ### Weakness 3
> From a theoretical standpoint, the conditions we impose are relatively mild compared to those in related works. Previous analyses either enforce strong structural constraints on weight parameters [1–4] or explicitly construct weights to satisfy certain properties [5–7]. While our Condition 3.1 does introduce some structural assumption, it is intended to illustrate at a high level the circumstances under which model weights can mitigate the influence of positional encodings. Assumption 3.1 only requires that two elements of  $Q^\top K$  and  V  be sufficiently close, which is a notably loose constraint. Furthermore, replacing this assumption with a general norm bound on the weight matrices would preserve our main conclusion regarding the \( O(k/N) \) error rate. We introduce Assumption 3.1 primarily to ensure the perturbation  $\Delta y_{N+1}$  remains minimal even for small values of  $N$ .
>
>
> ### Question 1
> The term of $\mathcal{O}(k/N)$ will still preserve for non-Gaussian data $x_i$ as long as the expectation of $\sum x_i^2$ is finite. For real-world datasets or open-source LLMs the situation becomes much more complicated and is perhaps beyond the scope of our paper.
>
> ### Question 2
> Please refer to the official commentary titled "Summary of our paper revision" for details regarding the extension to RoPE. Extending our analysis to the softmax activation function, however, presents significant challenges due to its inherent nonlinearity. This nonlinear nature precludes the direct derivation of meaningful, closed-form expressions for the prediction perturbation  $\Delta y_{N+1}$ . We suspect that generalizing our current results would require the activation function to possess certain linearity-like properties, such as $ \sigma(x + y) = \sigma(x) + \sigma(y) $, which softmax does not satisfy.
>
> ### Question 3
> As noted in our rebuttal to Weakness 3, Condition 3.1 is introduced primarily to offer high-level insight and to suggest the plausibility that such a pretrained model may exist. Assumption 3.1, however, is overly microscopic to visualize effectively, as it requires inspecting specific elements of a very large matrix. Such visualization would be highly sensitive to minor stochastic variations during training. Moreover, this assumption is not fundamental to our core result, as it could be relaxed or removed without altering the main theoretical conclusions.
>
> [1] Von Oswald, J., Niklasson, E., Randazzo, E., Sacramento, J., Mordvintsev, A., Zhmoginov, A., & Vladymyrov, M. (2023, July). Transformers learn in-context by gradient descent. In International Conference on Machine Learning (pp. 35151-35174). PMLR.
>
> [2] Ahn, K., Cheng, X., Daneshmand, H., & Sra, S. (2023). Transformers learn to implement preconditioned gradient descent for in-context learning. Advances in Neural Information Processing Systems, 36, 45614-45650.
>
> [3] Gatmiry, K., Saunshi, N., Reddi, S. J., Jegelka, S., & Kumar, S. (2024). Can looped transformers learn to implement multi-step gradient descent for in-context learning?. arXiv preprint arXiv:2410.08292.
>
> [4] Mahankali, A., Hashimoto, T. B., & Ma, T. (2023). One step of gradient descent is provably the optimal in-context learner with one layer of linear self-attention. arXiv preprint arXiv:2307.03576.
>
> [5] Bai, Y., Chen, F., Wang, H., Xiong, C., & Mei, S. (2023). Transformers as statisticians: Provable in-context learning with in-context algorithm selection. Advances in neural information processing systems, 36, 57125-57211.
>
> [6] Guo, T., Hu, W., Mei, S., Wang, H., Xiong, C., Savarese, S., & Bai, Y. (2023). How do transformers learn in-context beyond simple functions? a case study on learning with representations. arXiv preprint arXiv:2310.10616.
>
> [7] Wang, Z., Jiang, B., & Li, S. (2024, July). In-context learning on function classes unveiled for transformers. In Forty-first International Conference on Machine Learning.

---

### Author Response · Authors · 2025-11-21
**Summary of our paper revision**

We have introduced Section 3.2.2, which presents a new theorem (Theorem 3.2) concerning Rotary Positional Encoding (RoPE). Unlike our previous results, this theorem does not rely on Assumption 3.1. Our analysis reveals that the expected prediction perturbation under RoPE scales as $C_{\text{RoPE}} \cdot (k d^3 / N)$, where the constant $C_{\text{RoPE}}$ incorporates the maximum frequency parameter $\max_i \theta_i$ of the RoPE rotation matrix. To validate this theoretical finding, we have added corresponding experimental results in Figure 2. Additionally, we have included a brief preliminary overview of RoPE and removed the original Figure 1 to improve the paper’s focus and clarity.

---

### Meta-Review · Area_Chair_Fmnc · 2026-01-06

**Summary:**

This paper analyzes the impact of positional encoding on ICL performance from a theoretical perspective. If focuses on a linear transformer with one-hot positional encoding.

**Reviewer Concerns:**

- The analysis relies on simplified assumptions that are far away from transformers in practice.
- Theoretical results are largely constructive “existence” proofs, which are unsurprising and provide limited insight into whether or how real, trained transformers behave in practice.
- Experiments are synthetic with no evaluations on real-world sequential tasks. No ablations on modern architectures or positional encoding schemes.

The major concerns remain after rebuttal.

**Reviewer Scores:**

The original scores were 4/6/2/4/6. One reviewer (2) indicated that they would keep the score. The other reviewers would likely maintain their scores as well.

---

### Decision · Program_Chairs · 2026-01-26

Reject